# Online Reinforcement Learning
# for Mixed Policy Scopes

**Junzhe Zhang**
CausalAI Lab
Columbia University
junzhez@cs.columbia.edu

**Elias Bareinboim**
CausalAI Lab
Columbia University
eb@cs.columbia.edu

## Abstract

Combination therapy refers to the use of multiple treatments – such as surgery, medication, and behavioral therapy - to cure a single disease, and has become a cornerstone for treating various conditions including cancer, HIV, and depression. All possible combinations of treatments lead to a collection of treatment regimens (i.e., policies) with mixed scopes, or what physicians could observe and which actions they should take depending on the context. In this paper, we investigate the online reinforcement learning setting for optimizing the policy space with mixed scopes. In particular, we develop novel online algorithms that achieve sublinear regret compared to an optimal agent deployed in the environment. The regret bound has a dependency on the maximal cardinality of the induced state-action space associated with mixed scopes. We further introduce a canonical representation for an arbitrary subset of interventional distributions given a causal diagram, which leads to a non-trivial, minimal representation of the model parameters.

## 1  Introduction

The problem of policy learning is concerned with choosing actions based on the state of the environment with the goal of optimizing a certain measure of performance. Existing policy learning methods could be generally categorized into *online learning* and *offline learning*. Online reinforcement learning (RL) learns optimal policies by conducting sequential experimentation, while repeatedly adjusting the policy that is currently deployed based on the observed outcomes up to a certain point in time. Effective online algorithms have been developed for various canonical environments, including multi-armed bandits [26, 13, 1, 14], Markov decision processes (MDPs) [30, 11, 34], and partially observable MDP [9, 3]. On the other hand, off-policy learning focuses on the offline setting with the goal of evaluating the effectiveness of candidate policies from history data collected following a different behavior policy [24, 32, 33]. More generally, the discipline of causal inference (CI) offers focuses a compelling set of tools and a formal language for offline learning. It allows the agent to draw conclusions about new policies from a combination of observations and knowledge about the data-generating mechanisms. Several methods and graphical criteria have been proposed [23, 29, 4].

By and large, almost all methods described above concerns optimizing over a parametric space of policies with a fixed state-action space, which are called the *policy scope*. That is, the set of actions and observed states are pre-specified *a priori*. However, in many practical applications, this is possibly somewhat stringent, and the agent has to optimize over candidate policies with varying state-action spaces; these are called *mixed scopes*. For concreteness, consider the causal model in Fig. 1a that describes possible treatments for alcohol use disorder [19]. Based on the condition of alcohol dependant patients $Z$, the physician may prescribe a medication $X_1$ to maximize the total days of abstinence $Y$. An alternative treatment is to prescribe a behavioral therapy $X_2$, which alters the social environment $W$ of the patient, and, in turn, changes the pattern of alcohol use. The goal

36th Conference on Neural Information Processing Systems (NeurIPS 2022).

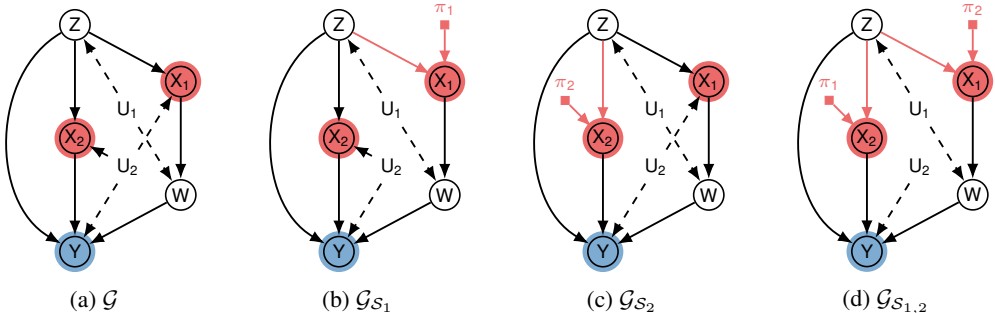

Figure 1: (a) A causal diagram $\mathcal{G}$; (b,c,d) policy-induced diagrams obtained from $\mathcal{G}$ associated with scopes $\mathcal{S}_1 = \{\langle X_1, \{Z\}\rangle\}$, $\mathcal{S}_2 = \{\langle X_2, \{Z\}\rangle\}$ and $\mathcal{S}_{1,2} = \{\langle X_1, \{Z\}\rangle, \langle X_2, \{Z\}\rangle\}$. Red nodes represent actions, blue nodes represent the reward, and red arrows for functional dependencies induced by interventions; $\pi$ are policy indicators, which will be left implicit throughout the paper.

of the analysis is to assess the effect of prescribing a combination treatment of both medication and behavioral therapy (i.e., a policy $\pi = \{\pi_1(X_1 \mid Z), \pi_2(X_2 \mid Z)\}$, compared to effects of prescribing each treatment separately (i.e., a policy $\pi = \{\pi_1(X_1 \mid Z)\}$ or $\pi = \{\pi_2(X_2 \mid Z)\}$). In this example, the action scope for all three treatment regimens are different, corresponding to $\{X_1, X_2\}$, $\{X_1\}$, and $\{X_2\}$, respectively. The physician has to determine the optimal set of actions to intervene in order to find the optimal policy so as to maximizing the days of abstinence.

There exists a growing literature concerned with the problem of optimizing mixed policies under various assumptions about the underlying environment. For instance, in the case of multi-armed bandits with linear reward functions, effective online algorithms exist for determining an optimal subset of actions so that intervening on them maximizes the expected reward [6, 5]. More recently, there exists new developments for policy optimization based on structural constraints in causal models. Non-parametric knowledge of causal relationships has been exploited to refine the space of policies with mixed scopes in increasingly relaxed scenarios [16, 17, 18]. In particular, there are now graphical conditions under which one could (1) identify necessary states for a set of actions, and (2) determine a partial ordering over policy scopes with regard to their maximum achievable expected rewards (optimality). These results allow an agent to detect and remove suboptimal and inefficient scopes, and focus its attention on the possibly optimal actions, accelerating the learning processes.

Despite of the the substantive progress achieved so far, an effective online algorithms for learning optimal policies with mixed scopes in an arbitrary causal system is still missing. In terms of graphical characterization, it is also highly non-trivial, and still unknown how the refinement of the candidate policy space affects the agent's performance. The goal of this paper is to address these challenges. We will investigate the online learning of optimal policies with mixed scopes, provided with a causal diagram associated with the underlying structural causal model (SCM) [23, Ch. 7]. In particular, our contributions are summarized as follows. (1) We develop a novel online learning algorithm that identifies an optimal policy with mixed scopes in an unknown SCM, and show that it achieves a sublinear cumulative regret. (2) We introduce a novel parametrization for SCMs with finite latent states, which can represent an arbitrary subset of interventional distributions through a minimal collection of c-components. (3) Leveraging these results, we develop an alternative online algorithm that is more computational efficient, while achieving the same asymptotic bound over the cumulative regret. Given the space constraints, all proofs are provided in the technical report [42, Appendix A].

## 1.1 Preliminaries

In this section, we introduce the basic notations and definitions used throughout the paper. We use capital letters to denote random variables ($X$), small letters for their values ($x$) and $\Omega_X$ for the domain of $X$. For an arbitrary set $\boldsymbol{X}$, let $|\boldsymbol{X}|$ be its cardinality. We denote by $P(\boldsymbol{X})$ represents a probability distribution over variables $\boldsymbol{X}$. Similarly, $P(\boldsymbol{Y} \mid \boldsymbol{X})$ represents a set of conditional distributions $P(\boldsymbol{Y} \mid \boldsymbol{X} = \boldsymbol{x})$ for all realizations $\boldsymbol{x}$. We will consistently use $P(\boldsymbol{x})$ as abbreviations for probabilities $P(\boldsymbol{X} = \boldsymbol{x})$; so does $P(\boldsymbol{Y} = \boldsymbol{y} \mid \boldsymbol{X} = \boldsymbol{x}) = P(\boldsymbol{y} \mid \boldsymbol{x})$. Finally, $\mathbb{1}\{\boldsymbol{Z} = \boldsymbol{z}\}$ is an indicator function that returns 1 if event $\boldsymbol{Z} = \boldsymbol{z}$ holds true; otherwise, it returns 0.

The basic semantical framework of our analysis rests on *structural causal models* (SCMs) [23, 4]. An SCM $M$ is a tuple $\langle \mathbf{V}, \mathbf{U}, \mathcal{F}, P(\mathbf{U}) \rangle$, where $\mathbf{V}$ is a set of endogenous variables and $\mathbf{U}$ is a set of exogenous variables. $\mathcal{F}$ is a set of functions s.t. each $f_V \in \mathcal{F}$ decides values of an endogenous variable $V \in \mathbf{V}$ taking as argument a combination of other variables in the system. That is, $V \leftarrow f_V(\mathbf{PA}_V, \mathbf{U}_V), \mathbf{PA}_V \subseteq \mathbf{V}, \mathbf{U}_V \subseteq \mathbf{U}$. Exogenous variables $U \in \mathbf{U}$ are mutually independent, values of which are drawn from the exogenous distribution $P(\mathbf{U})$. Naturally, $M$ induces a joint distribution $P(\mathbf{V})$ over endogenous variables $\mathbf{V}$, called the *observational distribution*.

A policy $\pi$ over a subset $\mathbf{X} \subseteq \mathbf{V}$ is a sequence of decision rules $\{\pi(X \mid \mathbf{S}_X)\}_{X \in \mathbf{X}}$, where every $\pi(X \mid \mathbf{S}_X)$ is a probability distribution mapping from domains of a set of covariates $\mathbf{S}_X \subseteq \mathbf{V}$ to the domain of an action $X$. An intervention following a policy $\pi$ over variables $\mathbf{X}$, denoted by $\mathrm{do}(\pi)$, is an operation which sets values of every $X \in \mathbf{X}$ to be decided by policy $X \sim \pi(X \mid \mathbf{S}_X)$ [37], replacing the functions $f_{\mathbf{X}} = \{f_X : \forall X \in \mathbf{X}\}$ that would normally determine their values. For an SCM $M$, let $M_\pi$ be a submodel of $M$ induced by intervention $\mathrm{do}(\pi)$. For a set $\mathbf{Y} \subseteq \mathbf{V}$, the interventional distribution $P(\mathbf{Y}|\mathrm{do}(\pi))$ is defined as the distribution over $\mathbf{Y}$ in the submodel $M_\pi$, i.e., $P_M(\mathbf{Y}|\mathrm{do}(\pi)) \triangleq P_{M_\pi}(\mathbf{Y})$. Subscript $M$ is left implicit when it is obvious from the context.

Each SCM $M$ is also associated with a causal diagram $\mathcal{G}$ (e.g., Fig. 1a), which is a directed acyclic graph (DAG) where solid nodes represent endogenous variables $\mathbf{V}$, empty nodes represent exogenous variables $\mathbf{U}$, and arrows represent the arguments $\mathbf{PA}_V, \mathbf{U}_V$ of each structural function $f_V$. We will use standard graph-theoretic family abbreviations to represent graphical relationships such as parents, children, descendants, and ancestors. For example, the set of parent nodes of $\mathbf{X}$ in $\mathcal{G}$ is denoted by $pa(\mathbf{X})_{\mathcal{G}} = \cup_{X \in \mathbf{X}} pa(X)_{\mathcal{G}}$; $ch$, $de$ and $an$ are similarly defined. Capitalized versions $Pa, Ch, De, An$ include the argument as well, e.g. $Pa(\mathbf{X})_{\mathcal{G}} = pa(\mathbf{X})_{\mathcal{G}} \cup \mathbf{X}$. For a subset $\mathbf{X} \subseteq \mathbf{V}$, $\mathcal{G}[\mathbf{X}]$ is a vertex-induced subgraph from $\mathcal{G}$, which contains nodes $\mathbf{X}$ and edges among them.

For convenience, we define a *bidirected arrow* $V_i \leftrightarrow V_j$ between endogenous nodes $V_i, V_j \in \mathbf{V}$ as a sequence of arrows $V_i \leftarrow U_k \rightarrow V_k$ where $U_k \in \mathbf{U}$ is an exogenous parent shared by $V_i, V_j$. A *bi-directed path* is a consecutive sequence of bi-directed arrows. We will utilize a special type of clustering of nodes in the diagram $\mathcal{G}$, called the *c-component* [38]. Formally, a subset $\mathbf{C} \subseteq \mathbf{V}$ is a c-component in a causal diagram $\mathcal{G}$ if any pair $V_i, V_j \in \mathbf{C}$ is connected by a bi-directed path in $\mathcal{G}$. For instance, there exist bidirected paths $X_1 \leftrightarrow X_2 \leftrightarrow Y$ and $Z \leftrightarrow W$ in the causal diagram $\mathcal{G}$ in Fig. 1a. $\mathcal{G}$ thus contains c-components $\{X_1, X_2, Y\}, \{Z, W\}$. On the other hand, the diagram $\mathcal{G}_{\mathcal{S}_1}$ in Fig. 1b contains c-components $\{X_1\}, \{X_2, Y\}, \{Z, W\}$, since bi-directed arrows $X_1 \leftrightarrow X_2$ and $Y \leftrightarrow X_1$ are removed. For a detailed survey on SCMs, we refer readers to [23, Ch. 7].

## 2 Optimizing Mixed Policy Spaces

We are concerned with the decision-making setting where an agent interacts with an SCM to optimize a reward $Y$. The agent could intervene on an arbitrary subset of actions $\mathbf{X}$, called *intervenable variables*. A *policy scope* $\mathcal{S}$ is a collections of pairs $\langle X, \mathbf{S}_X \rangle$ where $X \in \mathbf{X}$ and covariates $\mathbf{S}_X \subseteq \mathbf{V} \subseteq \{Y\}$. Let $\mathbf{X}(\mathcal{S}) = \{X\}_{\langle X, \mathbf{S}_X \rangle \in \mathcal{S}}$ denote the subset of actions in scope $\mathcal{S}$; similarly, $\mathbf{S}(\mathcal{S}) = \bigcup_{\langle X, \mathbf{S}_X \rangle \in \mathcal{S}} \mathbf{S}_X$. For instance, for a scope $\mathcal{S}_1 = \{\langle X_1, \{Z\} \rangle\}$ in Fig. 1b, $\mathbf{X}(\mathcal{S}_1) = \{X_1\}$ and $\mathbf{S}(\mathcal{S}_1) = \{Z\}$. Formally, a policy $\pi$ associated with scope $\mathcal{S}$ is a sequence of decision rules $\{\pi(X \mid \mathbf{S}_X)\}_{\langle X, \mathbf{S}_X \rangle \in \mathcal{S}}$. The collection of such policies $\pi$ define a *policy space* $\Pi_{\mathcal{S}}$. All policies in $\Pi_{\mathcal{S}}$ share the same scope $\mathcal{S}$: they intervene on all actions in $\mathbf{X}(\mathcal{S})$ based on observed values of covariates $\mathbf{S}(\mathcal{S})$. Policy space $\Pi_{\mathcal{S}}$ models a general class of decision rules and treatment regimens in many classic sequential decision making settings, including policies in Markov decision processes [25], strategy profiles in influence diagrams [12], and dynamic treatment regimes [20].

This paper investigates a more general space of policies that are not restricted to a single scope $\mathcal{S}$. A *mixed policy scope* is a combinatorial set of policy scopes over intervenable variables $\mathbf{X}$. Formally,

**Definition 1.** A *mixed policy scope* $\mathbb{S}$ is a collection of policy scopes $\mathcal{S}$ such that $\mathbf{X}(\mathcal{S}) \subseteq \mathbf{X}$.

A policy space $\Pi_{\mathbb{S}}$ with regard to a mixed policy scope $\mathbb{S}$ is the set of all policies compatible with every scope in $\mathbb{S}$, i.e., $\Pi_{\mathbb{S}} = \cup_{\mathcal{S} \in \mathbb{S}} \Pi_{\mathcal{S}}$. Henceforth, we will consistently refer to $\Pi_{\mathbb{S}}$ as a *mixed policy space* [18]. Our goal is to learn an optimal policy in $\Pi_{\mathbb{S}}$ that maximize reward $Y$ in an SCM $M$, i.e.,

$$\pi^* = \arg\max_{\pi \in \Pi_{\mathbb{S}}} \mathbb{E}_M[Y \mid \mathrm{do}(\pi)] \tag{1}$$

The agent does not have access to the parametrization of the underlying SCM $M$. Instead, the agent can observe endogenous variables $\boldsymbol{V}$, the mixed policy space $\Pi_{\mathbb{S}}$, and the causal diagram $\mathcal{G}$ representing qualitative knowledge encoded about $M$. Throughout this paper, we assume that endogenous variables $\boldsymbol{V}$ are discrete and finite, while exogenous variables $\boldsymbol{U}$ could take any (continuous) value. Distributions $P(\boldsymbol{V})$ and $P(\boldsymbol{V} \mid \text{do}(\pi))$ are thus categorical probability measures.

## 2.1 A Causal Upper Confidence Bound Algorithm

A typical online algorithm evaluates the effectiveness of candidate policies $\pi \in \Pi_{\mathbb{S}}$ by directly deploying policies in the actual environment (SCM $M$) through repeated rounds of interventions $t = 1, 2, \ldots, T$. For each round $t$, the algorithm selects a policy $\pi_t \in \Pi_{\mathbb{S}}$, performs an intervention $\text{do}(\pi_t)$, and receives a subsequent reward $Y_t$. The cumulative regret for an online algorithm in an SCM $M$ after $T$ rounds of interventions is defined as $R(T, M) = \sum_{t=1}^{T} (\mathbb{E}_M[Y \mid \text{do}(\pi^*)] - Y_t)$. In words, $R(T, M)$ measures the cumulative loss incurred since the online algorithm does not always selects the optimal policy $\pi^*$. A reasonable, or desirable property for an online algorithm is to have a sublinear regret for any SCM $M$, i.e., $\lim_{T \to \infty} R(T, M)/T = 0$. If the regret is sublinear, the agent is known to be selecting the optimal policy almost all of the time as $T$ goes to infinity.

We will introduce a novel online algorithm that optimizes the mixed policy space while achieving a sublinear regret. It follows the well-celebrated principle of optimism in the face of uncertainty. That is, for each episode, the agent picks a policy based on *upper confidence bounds* (UCB) [1, 2, 39, 40, 41], which evaluate the effectiveness of candidate policies in the most optimistic models, compatible with past observations, that induces the maximal expected reward. One innovation of our approach is to leverage the invariances embedded in the causal structure $\mathcal{G}$, which allows us to evaluate a candidate policy using samples induced by other policies with different scopes.

Formally, let $\mathbb{C}(\mathcal{G})$ denote a *c-collection* containing all (maximal) c-components in a causal diagram $\mathcal{G}$ [1]. Consequently, $\mathbb{C}(\mathcal{G})$ forms a partition $\{\boldsymbol{C}_1, \ldots, \boldsymbol{C}_n\}$ over nodes in $\mathcal{G}$ where there is not bi-directed arrow between every pair $\boldsymbol{C}_i$ and $\boldsymbol{C}_j$ whenever $i \neq j$. Consider again the causal diagram $\mathcal{G}$ in Fig. 1a, c-collection $\mathbb{C}(\mathcal{G})$ contains c-components $\boldsymbol{C}_1 = \{X_1, X_2, Y\}$, $\boldsymbol{C}_2 = \{Z, W\}$, which forms a partition over nodes in $\mathcal{G}$. For an arbitrary subset $\boldsymbol{C} \subseteq \boldsymbol{V}$, the c-factor $Q[\boldsymbol{C}]$ is a function defined as $Q[\boldsymbol{C}](\boldsymbol{v}) = P(\boldsymbol{c} \mid \text{do}(\boldsymbol{v} \setminus \boldsymbol{c}))$ [38], where $\text{do}(\boldsymbol{v} \setminus \boldsymbol{c})$ is an atomic intervention setting values of $\boldsymbol{V} \setminus \boldsymbol{C}$ to a constant $\boldsymbol{v} \setminus \boldsymbol{c}$ [23]. For convenience, we often omit input $\boldsymbol{v}$ and write $Q[\boldsymbol{C}]$. Fix a policy scope $\mathcal{S} \in \mathbb{S}$. For any policy $\pi \in \Pi_{\mathcal{S}}$, let set $\boldsymbol{Z} = An(Y)_{\mathcal{G}_{\mathcal{S}}} \setminus \boldsymbol{X}(\mathcal{S})$. The interventional distribution $P(Y \mid \text{do}(\pi))$ could be decomposed over c-components in subgraph $\mathcal{G}[\boldsymbol{Z}]$ as follows:

$$P(y \mid \text{do}(\pi)) = \sum_{\boldsymbol{z}} \prod_{X \in \boldsymbol{X}(\mathcal{S})} \pi(x \mid \boldsymbol{s}_X) \prod_{\boldsymbol{C} \in \mathbb{C}_{\mathcal{S}}} Q[\boldsymbol{C}], \tag{2}$$

where the c-collection $\mathbb{C}_{\mathcal{S}} = \mathbb{C}(\mathcal{G}[\boldsymbol{Z}])$ contains all c-components in $\mathcal{G}[\boldsymbol{Z}]$. Let a c-collection $\mathbb{C} = \cup_{\mathcal{S} \in \mathbb{S}} \mathbb{C}_{\mathcal{S}}$. Among quantities in the above equation, decision rules $\pi(x \mid \boldsymbol{s}_X)$ are fixed. To evaluate effects $P(Y \mid \text{do}(\pi))$ of candidate policies in $\Pi_{\mathbb{S}}$, it is thus sufficient to learn parameters of $Q[\boldsymbol{C}]$ for components $\boldsymbol{C} \in \mathbb{C}$. Consider again the causal diagram $\mathcal{G}$ in Fig. 1a; the mixed policy scope $\mathbb{S}$ contains elements $\mathcal{S}_1 = \{\langle X_1, \{Z\}\rangle\}$, $\mathcal{S}_2 = \{\langle X_2, \{Z\}\rangle\}$ and $\mathcal{S}_{1,2} = \{\langle X_1, \{Z\}\rangle, \langle X_2, \{X_1, Z\}\rangle\}$. For a policy $\pi_1(x_1|z)$ characterized with scope $\mathcal{S}_1 = \{\langle X_1, \{Z\}\rangle\}$, $P(Y|\text{do}(\pi_1))$ decomposes as:

$$P(y \mid \text{do}(\pi_1)) = \sum_{x_1, x_2, z, w} \pi(x_1 \mid z) Q[X_2, Y] Q[Z, W]. \tag{3}$$

The c-collection $\mathbb{C}_{\mathcal{S}_1} = \{\{X_2, Y\}, \{Z, W\}\}$. Similarly, enumerating c-components associated with every scope $\mathcal{S} \in \mathbb{S}$ gives a c-collection $\mathbb{C} = \{\{Z, W\}, \{X_1, Y\}, \{X_2, Y\}, \{Y\}\}$. For any $\boldsymbol{C} \in \mathbb{C}$, let $\mathbb{S}(\boldsymbol{C})$ be a subset of scopes $\mathcal{S} \in \mathbb{S}$ such that $\boldsymbol{C} \in \mathbb{C}_{\mathcal{S}}$. Our online algorithm evaluates $Q[\boldsymbol{C}]$ by directly performing intervention $\text{do}(\pi)$ in the actual SCM $M$.

**Lemma 1.** *For a causal diagram $\mathcal{G}$, let $\boldsymbol{C} \in \mathbb{C}$ be a c-component in $\mathcal{G}$. Then, c-factor $Q[\boldsymbol{C}]$ factorizes over a topological ordering $\prec$ in $\mathcal{G}$ as follows:*

$$Q[\boldsymbol{C}] = \prod_{V \in \boldsymbol{C}} q\left(v \mid \boldsymbol{pa}_V^+\right) \tag{6}$$

*where extended parents $\boldsymbol{PA}_V^+ = Pa(\boldsymbol{C}_V) \setminus \{V\}$; $\boldsymbol{C}_V$ is the c-component containing $V$ in $\mathcal{G}[\{V' \in \boldsymbol{C} \mid V' \prec V\}]$. Moreover, $q\left(V \mid \boldsymbol{PA}_V^+\right) = P\left(V \mid \boldsymbol{PA}_V^+, do(\pi)\right)$ for any policy $\pi \in \Pi_{\mathbb{S}(\boldsymbol{C})}$.*

---

[1]Generally, a c-collection $\mathbb{C}$ is a collection of c-components $\boldsymbol{C}$ contain in a causal diagram $\mathcal{G}$.

**Algorithm 1** CAUSAL-UCB*

**Input:** Causal diagram $\mathcal{G}$, policy space $\Pi_{\mathbb{S}}$, failure tolerance $\delta \in (0, 1)$.
1: **for all** episodes $t = 1, 2, \ldots$ **do**
2:    For every $\boldsymbol{C} \in \mathbb{C}$, compute an empirical estimate $\hat{Q}_t[\boldsymbol{C}]$ following Eq. (8).
3:    Let $\mathscr{M}_t$ be a set of all SCMs $M$ associated with $\mathcal{G}$ such that each c-factor $Q_M[\boldsymbol{C}]$ is close to its estimate $\hat{Q}_t[\boldsymbol{C}]$ in Eq. (8). That is, for every $V \in \boldsymbol{C}$, for any $\boldsymbol{pa}_V^+$,

$$\left\| q\left( \cdot \mid \boldsymbol{pa}_V^+ \right) - \hat{q}_t\left( \cdot \mid \boldsymbol{pa}_V^+ \right) \right\|_1 \leq \sqrt{\frac{6 |\Omega_V| \ln\left( 2 |\Omega_{\boldsymbol{PA}_V^+}| |\boldsymbol{V}(\mathbb{C})| t/\delta \right)}{\max\left\{ n_t\left( \boldsymbol{pa}_V^+ \right), 1 \right\}}}. \tag{4}$$

4:    Find the optimistic policy $\boldsymbol{\pi}_t$ such that

$$\boldsymbol{\pi}_t = \arg\max_{\boldsymbol{\pi} \in \Pi_{\mathbb{S}}} \max_{M \in \mathscr{M}_t} \mathbb{E}_M[Y \mid \mathrm{do}(\boldsymbol{\pi})] \tag{5}$$

5:    Perform $do\left( \boldsymbol{\pi}_t \right)$ and observe $\boldsymbol{V}^{(t)}$.

As an example, consider again the causal diagram $\mathcal{G}$ and policy scopes $\mathcal{S}_1, \mathcal{S}_2, \mathcal{S}_{1,2}$ described in Fig. 1. For a c-component $\boldsymbol{C}_3 = \{X_2, Y\} \in \mathbb{C}_{\mathcal{S}_1}$, this implies the policy scope $\mathcal{S}_1 \in \mathbb{S}(\boldsymbol{C}_3)$. Observe that $\boldsymbol{C}_Y = \{X_2, Y\}$, $\boldsymbol{C}_{X_2} = \{X_2\}$ and extended parents $\boldsymbol{PA}_Y^+ = \{X_2, Z, W\}$, $\boldsymbol{PA}_{X_2}^+ = \{Z\}$. Lem. 1 implies $Q[\boldsymbol{C}_3]$ is computable from the sampling process following any policy $\pi_1 \in \Pi_{\mathcal{S}_1}$, i.e.,

$$Q[X_2, Y] = P(y \mid x_2, z, w, \mathrm{do}(\pi_1)) P(x_2 \mid z, \mathrm{do}(\pi_1)). \tag{7}$$

Details of our proposed algorithm, CAUSAL-UCB*, are provided in Alg. 1. It interacts with the underlying SCM through policies in $\Pi_{\mathbb{S}}$ in repeated episodes $t = 1, \ldots, T$. At episode $t$, it computes an empirical estimate of c-factor $Q[\boldsymbol{C}]$ for each c-component $\boldsymbol{C} \in \mathbb{C}$. More specifically, the empirical estimate of c-factor $\hat{Q}_t[\boldsymbol{C}]$ prior to episode $t$ is given by:

$$\hat{Q}_t[\boldsymbol{C}](\boldsymbol{v}) = \prod_{V \in \boldsymbol{C}} \hat{q}_t\left( v \mid \boldsymbol{pa}_V^+ \right), \text{ where } \hat{q}_t\left( v \mid \boldsymbol{pa}_V^+ \right) = \frac{n_t\left( v, \boldsymbol{pa}_V^+ \right)}{\max\left\{ n_t\left( \boldsymbol{pa}_V^+ \right), 1 \right\}}. \tag{8}$$

Among quantities in the above equation, $n_t\left( v, \boldsymbol{pa}_V^+ \right)$ is the event count for observing $V = v$, $\boldsymbol{PA}_V^+ = \boldsymbol{pa}_V^+$ after deploying policy $\pi \in \Pi_{\mathbb{S}(\boldsymbol{C})}$ prior to episode $t$. That is, $n_t\left( v, \boldsymbol{pa}_V^+ \right) = \sum_{\tau=1}^{t-1} \mathbb{1}\left\{ V_\tau = v, \left( \boldsymbol{PA}_V^+ \right)_\tau = \boldsymbol{pa}_V^+, \boldsymbol{\pi}_\tau \in \Pi_{\mathbb{S}(\boldsymbol{C})} \right\}$ and $n_t\left( \boldsymbol{pa}_V^+ \right) = \sum_v n_t\left( v, \boldsymbol{pa}_V^+ \right)$. At Step 3, CAUSAL-UCB* maintains a confidence set $\mathscr{M}_t$ of all possible SCM compatible with $\mathcal{G}$ using convex intervals centered around each estimate $\hat{Q}_t[\boldsymbol{C}]$. It then finds the optimal policy $\pi_t$ for the most optimistic instance $M_t$ from $\mathscr{M}_t$ that induces the maximal expected reward $\mathbb{E}[Y \mid \mathrm{do}(\pi)]$ (Step 4). Finally, Step 7 performs $\mathrm{do}(\pi_t)$ throughout episode $t$ and collect new samples $\boldsymbol{V}_t$. Let $\boldsymbol{V}(\mathbb{C})$ be a union $\bigcup_{\boldsymbol{C} \in \mathbb{C}} \boldsymbol{C}$. It is possible to derive a regret bound of CAUSAL-UCB* after $T > 1$ episodes.

**Theorem 1.** *For a causal diagram $\mathcal{G}$ and a mixed policy scope $\mathbb{S}$, fix a $\delta \in (0, 1)$. With probability at least $1 - \delta$, it holds for any $T > 1$, the regret of CAUSAL-UCB* is bounded by*

$$R(T, M) \leq 19 \Delta(\mathcal{G}, \mathbb{S}) \sqrt{|\mathbb{S}| T \ln\left( |\boldsymbol{V}(\mathbb{C})| T/\delta \right)}. \tag{9}$$

*where function $\Delta(\mathcal{G}, \mathbb{S}) = \max_{\mathcal{S} \in \mathbb{S}} \Delta(\mathcal{G}, \mathcal{S})$ and $\Delta(\mathcal{G}, \mathcal{S}) = \sum_{V \in \boldsymbol{V}(\mathbb{C}_{\mathcal{S}})} \sqrt{|\Omega_{V \cup \boldsymbol{PA}_V^+}|}$.*

Thm. 1 implies that CAUSAL-UCB* is able to achieve a sublinear regret; therefore, policy $\pi_t$ eventually converges to an optimal policy as episode $t \to \infty$. For example, in the causal diagram $\mathcal{G}$ and the mixed policy scope $\mathbb{S} = \{\mathcal{S}_1, \mathcal{S}_2, \mathcal{S}_{1,2}\}$ described in of Fig. 1, applying Thm. 1 gives a regret bound $\mathcal{O}\left( \Delta(\mathcal{G}, \mathbb{S}) \sqrt{T \ln T} \right)$. Evaluating $\Delta(\mathcal{G}, \mathbb{S})$ with the cardinality $|\Omega_{X_2}| > |\Omega_{X_1}|$ gives

$$\Delta(\mathcal{G}, \mathbb{S}) = \Delta(\mathcal{G}, \mathcal{S}_1) = \sqrt{|\Omega_{Y, X_2, Z, W}|} + \sqrt{|\Omega_{X_2, Z}|} + \sqrt{|\Omega_{W, Z, X_1}|} + \sqrt{|\Omega_Z|}. \tag{10}$$

In comparison, applying standard UCB with deterministic polices in $\Pi_{\mathbb{S}}$ as arms leads to a regret bound $\mathcal{O}\left( \sqrt{|\Pi_{\mathbb{S}}| T \ln T} \right)$ where $|\Pi_{\mathbb{S}}| = |\Omega_{X_1}|^{|\Omega_Z|} + |\Omega_{X_2}|^{|\Omega_Z|} + |\Omega_{X_1}|^{|\Omega_Z|} \times |\Omega_{X_2}|^{|\Omega_Z|}$, which is dominated by that of CAUSAL-UCB* as the number of endogenous states increases.

**Accelerate Learning Processes** In the regret bound of Thm. 1, function $\Delta(\mathcal{G}, \mathbb{S})$ can be seen as a measure evaluating the difficulty of the learning task. It has a dependency on the maximal cardinality of the state-action space associated with policy scopes $\mathcal{S}$ in $\mathbb{S}$. There exist a general characterization of the mixed scope with respect to properties that allow the agent to detect redundant and suboptimal policy scopes [18]. Doing so, we obtain a refinement of the agent's candidate policy scopes so that it converges to the optimal strategy faster and more robustly. For instance, consider the causal diagram $\mathcal{G}$

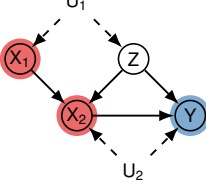

Figure 2

described in Fig. 2. We are interested in evaluating policies with mixed scope $\mathbb{S} = \{\mathcal{S}_1, \mathcal{S}_2, \mathcal{S}_{1,2}\}$ where $\mathcal{S}_1 = \{\langle X_1, \emptyset \rangle\}$, $\mathcal{S}_2 = \{\langle X_2, \emptyset \rangle\}$ and $\mathcal{S}_{1,2} = \{\langle X_1, \emptyset \rangle, \langle X_2, \{X_1\} \rangle\}$. It is possible to show that $\mathcal{S}_{1,2}$ is redundant since there always exists an optimal policy with scope $\mathcal{S}_1$ or $\mathcal{S}_2$. Removing $\mathcal{S}_{1,2}$ leads to a more refined mixes scope $\mathbb{S}^* = \{\mathcal{S}_1, \mathcal{S}_2\}$. Since $\Delta(\mathcal{G}, \mathbb{S}^*) = \max\{\Delta(\mathcal{G}, \mathcal{S}_1), \Delta(\mathcal{G}, \mathcal{S}_1)\} \leq \max\{\Delta(\mathcal{G}, \mathcal{S}_1), \Delta(\mathcal{G}, \mathcal{S}_1), \Delta(\mathcal{G}, \mathcal{S}_{1,2})\} = \Delta(\mathcal{G}, \mathbb{S})$, applying CAUSAL-UCB* with the refined scope $\mathbb{S}^*$ achieve a smaller regret than that with $\mathbb{S}$, thus accelerating the online learning process.

## 3 Optimistic Planning over Mixed Policy Scopes

The optimization problem entailed by CAUSAL-UCB* in the algorithm discussed earlier (Eq. (5)) requires the learner to search over all possible SCMs compatible with the causal diagram and experimental data. In principle, this entails a challenging task since one does not have access to the parametric forms of the underlying structural functions $\mathcal{F}$ nor the exogenous distribution $P(\boldsymbol{U})$. Consequently, it is not clear how the existing optimization procedures can be used.

When domains of endogenous variables are discrete and finite, there exists a canonical family of SCMs that parametrizes all observational and interventional distributions in any causal diagram using a finite number of exogenous states [43, 27]. A c-component $\boldsymbol{C} \in \mathbb{C}$ is said to cover an exogenous variable $U \in \boldsymbol{U}$ if $U \in \cup_{V \in \boldsymbol{C}} \boldsymbol{U}_V$. Let $\mathbb{C}(U)$ denote the set of c-components in $\mathbb{C}$ covering $U$. Consider again the causal diagram in Fig. 1a. For the c-collection $\mathbb{C} = \{\{Z, W\}, \{X_1, Y\}, \{X_2, Y\}, \{Y\}\}$, exogenous node $U_2$ is covered by $\mathbb{C}(U_2) = \{\{X_1, Y\}, \{X_2, Y\}, \{Y\}\}$; similarly, $\mathbb{C}(U_1) = \{\{Z, W\}\}$. Next we present a novel finite-state decomposition representing c-factors in any c-collection $\mathbb{C}$.

**Theorem 2.** *For any SCM $M = \langle \boldsymbol{V}, \boldsymbol{U}, \mathscr{F}, P(\boldsymbol{U}) \rangle$, let $\mathbb{C}$ be an arbitrary c-collection. For any $\boldsymbol{C} \in \mathbb{C}$, c-factor $Q[\boldsymbol{C}]$ decomposes as follows:*

$$Q[\boldsymbol{C}](\boldsymbol{v}) = \sum_{U \in \boldsymbol{U}} \sum_{u=1,\dots,d_U} \prod_{V \in \boldsymbol{C}} \mathbb{1}\{f_V(\boldsymbol{pa}_V, \boldsymbol{u}_V) = v\} \prod_{U \in \boldsymbol{U}} P(u) \quad (11)$$

*where for every exogenous $U \in \boldsymbol{U}$, $P(U)$ is a discrete distribution over a finite domain $\{1, \dots, d_U\}$ with cardinality $d_U = \sum_{\boldsymbol{C} \in \mathbb{C}(U)} |\Omega_{Pa(\boldsymbol{C})}|$; $\mathbb{C}(U) \subseteq \mathbb{C}$ are c-components covering $U$.*

Henceforth, we will refer to the family of SCMs with finite exogenous domain defined in Thm. 2 as $\mathbb{C}$-*canonical SCMs*. For concreteness, consider the causal diagram in Fig. 1a with $X_1, X_2, Y, Z, W \in \{0, 1\}$. Recall that for $\mathbb{C} = \{\{Z, W\}, \{X_1, Y\}, \{X_2, Y\}, \{Y\}\}$, $\mathbb{C}(U_1) = \{\{Z, W\}\}$ and $\mathbb{C}(U_2) = \{\{X_1, Y\}, \{X_2, Y\}, \{Y\}\}$. The cardinality of $U_1$ in $\mathbb{C}$-canonical SCMs is $d_1 = |\Omega_{Pa(Z,W)}| = |\Omega_{Z,X_1,W}| = 8$. Similarly, the cardinality of $U_2$ in $\mathbb{C}$-canonical SCMs is given by:

$$d_2 = |\Omega_{Pa(X_1,Y)}| + |\Omega_{Pa(X_2,Y)}| + |\Omega_{Pa(Y)}| = 2^5 + 2^4 + 2^4 = 64. \quad (12)$$

Thm. 2 implies that $Q[Z, W], Q[X_2, Y]$ in the diagram of Fig. 1a could be written as:

$$Q[Z, W] = \sum_{u_1=1,\dots,d_1} \mathbb{1}\{f_Z(u_1) = z\}\mathbb{1}\{f_W(z, x_1, u_1) = w\}P(u_1) \quad (13)$$

$$Q[X_2, Y] = \sum_{u_2=1,\dots,d_2} \mathbb{1}\{f_{X_2}(z, u_2) = x_2\}\mathbb{1}\{f_Y(x_2, z, w, u_2) = y\}P(u_2) \quad (14)$$

where $P(U_1), P(U_2)$ are distributions over finite domains $\{1, \dots, d_1\}, \{1, \dots, d_2\}$, respectively. Following the decomposition in Eq. (3), for any policy $\pi_1 \in \Pi_{\mathcal{S}_1}$, the interventional distribution $P(Y \mid \text{do}(\pi_1))$ is a function of c-factors $Q[X_2, Y], Q[Z, W]$ and is given by:

$$P(y \mid \text{do}(\pi_1)) = \sum_{x_1,x_2,z,w} \pi(x_1|z) \sum_{u_1=1,\dots,d_1} \mathbb{1}\{f_Z(u_1) = z\}\mathbb{1}\{f_W(z, x_1, u_1)\}P(u_1)$$
$$\sum_{u_2=1,\dots,d_2} \mathbb{1}\{f_{X_2}(z, u_2) = x_2\}\mathbb{1}\{f_Y(x_2, z, w, u_2) = y\}P(u_2) \quad (15)$$

Observe in Thm. 2, cardinalities of exogenous domains rely on the total number of c-components in $\mathbb{C}$. Next we introduce an effective method to reduce the model complexity of canonical SCMs by exploring identifiable relationships among c-factors $Q[C]$ contained in $\mathbb{C}$.

**Definition 2.** For a causal diagram $\mathcal{G}$ and a c-collection $\mathbb{C}$, c-factor $Q[C]$, $\forall C \in \mathbb{C}$, is identifiable if $Q[C]$ is uniquely computable from other c-factors $Q[C']$ in $\mathbb{C} \setminus \{C\}$. That is, $Q_{M_1}[C] = Q_{M_2}[C]$ for every pair of SCMs $M_1, M_2$ with $\mathcal{G}_{M_1} = \mathcal{G}_{M_2} = \mathcal{G}$, and $Q_{M_1}[C'] = Q_{M_2}[C']$ for any $C' \in \mathbb{C} \setminus \{C\}$.

In words, $Q[C]$ is identifiable w.r.t. $\langle \mathcal{G}, \mathbb{C} \rangle$ if it could be written as a function of c-factors in the remainder of the collection $\mathbb{C}' = \mathbb{C} \setminus \{C\}$. For instance, consider the the causal diagram of Fig. 1a and the c-collection $\mathbb{C} = \{\{Z, W\}, \{X_1, Y\}, \{X_2, Y\}, \{Y\}\}$. [36, Lem. 10] implies that c-factor $Q[Y]$ is identifiable from $Q[X_1, Y]$ and is given by $Q[Y] = \sum_{x_1} Q[X_1, Y]$. For every pair of SCMs that generates the same set of c-factors in $\mathbb{C}' = \{\{Z, W\}, \{X_1, Y\}, \{X_2, Y\}\}$, they must also coincide in parameters of $Q[Y]$. It is thus sufficient to focus on representing parameters of c-factors in the subset $\mathbb{C}'$, which we call a *reduction* of the original c-collection $\mathbb{C}$.

**Definition 3.** For a causal diagram $\mathcal{G}$ and a c-collection $\mathbb{C}$, a subset $\mathbb{C}' \subseteq \mathbb{C}$ is said to be a *reduction* of $\mathbb{C}$ if it is obtained by successive removals of identifiable c-component $C$.

A reduction $\mathbb{C}'$ is *minimal* if there exists no proper subset $\mathbb{C}'' \subset \mathbb{C}'$ such that $\mathbb{C}''$ is a reduction of $\mathbb{C}'$. To reduce the model complexity of candidate SCMs, it is always preferable to present c-components in a minimal reduction. Next we describe a systematic procedure to obtain a minimal reduction of c-collection $\mathbb{C}$ in a causal diagram $\mathcal{G}$. Details of our algorithm, MINCOLLECT, are described in Alg. 2. It repeatedly removes identifiable c-factors $Q[C]$ from collection $\mathbb{C}$. The subroutine IDENTIFY [36] is a complete algorithm in determining the identi-

---

**Algorithm 2** MINCOLLECT

**Input:** Causal diagram $\mathcal{G}$, c-collection $\mathbb{C}$.
1: **function** MINCOLLECT($\mathcal{G}, \mathbb{C}$)
2:     **while** $C = $ FINDID($\mathcal{G}, \mathbb{C}$) $\neq \emptyset$ **do**
3:         Let $\mathbb{C} = \mathbb{C} \setminus \{C\}$.
4:     **return** $\mathbb{C}$.
5: **function** FINDID($\mathcal{G}, \mathbb{C}$)
6:     **for** every pair $C, C' \in \mathbb{C}$ s.t. $C \subset C'$ **do**
7:         **if** IDENTIFY($C, C', \mathcal{G}$) $\neq$ FAIL **then**
8:             **return** $C$.

---

fiability of $Q[C]$ from another c-factor $Q[C']$ such that $C \subseteq C'$. In particular, IDENTIFY returns a formula estimating $Q[C]$ from $Q[C']$ if the target query is identifiable; otherwise, it returns "FAIL". It is possible to show that the greedy procedure in Alg. 2 always returns a minimal reduction of $\mathbb{C}$.

**Proposition 1.** *For a causal diagram $\mathcal{G}$ and a c-collection $\mathbb{C}$,* MINCOLLECT($\mathcal{G}, \mathbb{C}$) *returns a minimal reduction $\mathbb{C}^*$ of $\mathbb{C}$.*

A natural question arising at this point is whether the order of removing c-components $C$ could affect the output of MINCOLLECT. Fortunately, the next proposition shows that this is not the case.

**Proposition 2.** *For a causal diagram $\mathcal{G}$, any c-collection $\mathbb{C}$ has a unique minimal reduction.*

Consider again the c-collection $\mathbb{C} = \{\{Z, W\}, \{X_1, Y\}, \{X_2, Y\}, \{Y\}\}$ in the causal diagram $\mathcal{G}$ of Fig. 1a. Applying procedure MINCOLLECT($\mathbb{C}, \mathcal{G}$) gives a minimal reduction $\mathbb{C}^* = \{\{Z, W\}, \{X_1, Y\}, \{X_2, Y\}\}$. Therefore, it is sufficient to consider $\mathbb{C}^*$-canonical SCMs compatible wtih $\mathcal{G}$ when computing the optimistic policy in Eq. (5). Compared with $\mathbb{C}$-canonical SCMs, the cardinality of $U_1$ in a $\mathbb{C}^*$-canonical SCM remains the same and equates to $d_1 = |\Omega_{Pa(Z,W)}| = 8$. On the other hand, $\mathbb{C}^*(U_2) = \{\{X_1, Y\}, \{X_2, Y\}\}$ and the cardinality of $U_2$ is given by

$$d_2 = |\Omega_{Pa(X_1,Y)}| + |\Omega_{Pa(X_2,Y)}| = 2^5 + 2^4 = 48, \tag{16}$$

which is smaller than the cardinality of $U_2$ in $\mathbb{C}$-canonical SCMs given by Eq. (12). One could then obtain an optimistic policy in Eq. (5) by solving a series of equivalent polynomial programs. For a more detailed survey on canonical SCMs and related work, we refer readers to the complete technical report [42, Appendix C].

## 3.1 Thompson Sampling

The canonical representation of c-factors allows the optimization problem in Eq. (5) to be reducible to a series of equivalent polynomial programs. Nevertheless, solving polynomial optimization is NP-hard in general [8], which means that applying CAUSAL-UCB* is still computationally challenging. This section introduces an alternative online algorithm that is computationally feasible, while achieving a similar asymptotic bound on the cumulative regret.

Our algorithm is based on the heuristics of Thompson sampling (TS) [35, 31, 21]. It maintains a prior distribution $\rho(M)$ over all possible SCMs $M$ compatible with the causal diagram $\mathcal{G}$. One may surmise that it is challenging to define such a prior $\rho$ since the agent does not have access to the parametric forms of the underlying SCM. Fortunately, it follows from the decomposition in Thm. 2 that we could assume the exogenous domain to be discrete and finite without loss of generality. Particularly, we will consider a family of $\mathbb{C}^*$-canonical SCMs where $\mathbb{C}^*$ is a minimal reduction of c-collection $\mathbb{C}$ in $\mathcal{G}$. The cardinality of domain of every $U \in \boldsymbol{U}$ is given by $d_U = \sum_{\boldsymbol{C} \in \mathbb{C}^*(U)} |\Omega_{Pa(\boldsymbol{C})}|$. We assume that probabilities of $P(\boldsymbol{U})$ are drawn from uninformative Dirichlet priors; and $\mathcal{F}$ are drawn uniformly from the finite class of possible structural functions. That is, for every $U \in \boldsymbol{U}$ and every $V \in \boldsymbol{V}$,

$$P(U) \sim \texttt{Dir}\left(\alpha_1, \ldots, \alpha_{d_U}\right), \qquad f_V \sim \texttt{Unif}\left(\Omega_{\boldsymbol{PA}_V} \times \Omega_{\boldsymbol{U}_V} \mapsto \Omega_V\right), \qquad (17)$$

where $\alpha_1 = \cdots = \alpha_{d_U} = 1$; $\Omega_{\boldsymbol{PA}_V} \times \Omega_{\boldsymbol{U}_V} \mapsto \Omega_V$ is a class containing all functions mapping from finite domains of $\boldsymbol{PA}_V, \boldsymbol{U}_V$ to $V$, which must also contain only a finite number of elements.

Details of our algorithm, CAUSAL-TS*, are described in Alg. 3. At each episode $t$, it updates the posterior distribution $\rho(M|\bar{V}_t)$ from collected samples $\bar{V}_t = \{V_1, \ldots, V_{t-1}\}$ prior to episode $t$, and draws an estimate $M_t$ from the updated posteriors. Similar to Bayesian causal inference approaches [7, 43], we will obtain a posterior sample $M \sim \rho(M \mid \bar{V}_t)$ using Gibbs sampling [10]. In Step 4, CAUSAL-TS* computes an optimistic policy $\pi_t$ that maximizes the expected outcome $\mathbb{E}[Y \mid do(\pi)]$ induced by the sampled SCM $M_t$. Finally, the agent executes $\pi_t$ throughout episode $t$ and new samples $\boldsymbol{V}_t$ are collected.

---

**Algorithm 3** CAUSAL-TS*

1: **Input:** Causal diagram $\mathcal{G}$, policy space $\Pi_{\mathbb{S}}$, prior $\rho$.
2: **for all** episodes $t = 1, 2, \ldots$ **do**
3:     Sample an SCM $M_t \sim \rho\left(M \mid \bar{V}_t\right)$.
4:     Compute an optimal policy $\boldsymbol{\pi}_t$ such that

$$\boldsymbol{\pi}_t = \arg\max_{\boldsymbol{\pi} \in \Pi_{\mathbb{S}}} \mathbb{E}_{M_t}[Y \mid do(\boldsymbol{\pi})] \qquad (18)$$

5:     Perform $do\left(\boldsymbol{\pi}_t\right)$ and observe $\boldsymbol{V}^{(t)}$.

---

Following previous work [28, 21, 22], we will assess the performance of TS using the Bayesian cumulative regret up to episode $T$, i.e., $R(T, \rho) = \mathbb{E}\left[R(T, M) \mid M \sim \rho(M)\right]$, where the expectation is taken with respect to the prior distribution over $M$. There exists a general relationship between TS and UCB algorithms in many model classes [28]. This allows one to convert regret bounds developed for CAUSAL-UCB* into Bayesian regret bounds for CAUSAL-TS*. Formally,

**Theorem 3.** *Given a causal diagram $\mathcal{G}$, a mixed policy scope $\mathbb{S}$, and a prior distribution $\rho$, it holds for any $T > 1$, the regret of* CAUSAL-TS* *is bounded by*

$$R(T, \rho) \leq 26\Delta(\mathcal{G}, \mathbb{S})\sqrt{|\mathbb{S}|T \ln\left(|\boldsymbol{V}(\mathbb{C})|T\right)}. \qquad (19)$$

Compared with Thm. 1, the above regret bound implies that CAUSAL-TS* achieves a similar asymptotic performance as CAUSAL-UCB*. Particularly, Alg. 3 requires one only has to find an optimal policy $\pi_t$ in a specific SCM $M_t$, while in Alg. 1, parameters of $M_t$ are imprecise, bounded in the hypothesis class $\mathscr{M}_t$. There exist effective planning algorithms in finding optimal policies in a structured environment provided with detailed parameterization of the underlying SCM [15, 12]. This implies that CAUSAL-TS* is more computationally feasible compared to CAUSAL-UCB*.

## 4 Simulations

In this section, we evaluate the performance of our algorithms on randomly generated SCMs in various types of causal diagrams. After all, our algorithms can consistently find the corresponding optimal policies with mixed scopes. Further leveraging causal relationships in the underlying environment accelerates the convergence rate of online learners. In all experiments, we evaluate the novel CAUSAL-TS*, with uninformative Dirichlet priors over exogenous probabilities and uniform priors over structural functions, which we label as *c-ts*. As a baseline, we also include randomized trials (*rct*) allocating treatments in all possible scopes uniformly at random, standard Thompson sampling algorithm (*ts*) using all deterministic policies as arms, and Thompson sampling over a simplified mixed scope (*ts**), which is obtained by applying graphical conditions in [18]. For all algorithms, we measure their cumulative regrets over $T = 1.1 \times 10^3$ episodes. We refer readers to the technical report [42, Appendix B] for a more detailed discussion on the experimental set-up.

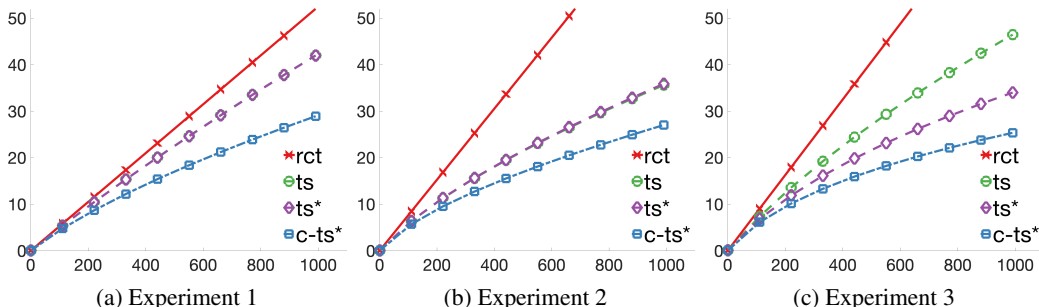

| | (a) Experiment 1 | (b) Experiment 2 | (c) Experiment 3 |

Figure 3: Simulations comparing online learners that are randomized (*rct*), adaptive (*ts*), adaptive with simplified policy scopes (*ts\**), and causally enhances (*c-ts\**). $x$-axle represents total episodes and $y$-axle for the cumulative regret. Figures are rendered in high resolution and can be zoomed in.

**Experiment 1**  Consider again the causal diagram and policy scopes described in Fig. 1. We randomly generate 100 instances of SCMs in Fig. 1a with binary $X_1, X_2, Z, W, Y \in \{0, 1\}$. Exogenous variables $U_1, U_2$ are discrete, taking values in finite domains with cardinalities $d_1 = 16, d_2 = 48$ respectively. Thm. 2 and Prop. 1 implies that this parametric family is sufficient to generate all possible effected reward of candidate policies in Fig. 1. The cumulative regrets averaging over random SCMs are reported in Fig. 3a. Our analysis reveals that *c-ts\** consistently outperforms *ts* and *ts\** since it is able to reuse samples induced by policies with mixed scopes by exploiting underlying causal knowledge. The performance of *ts* and *ts\** coincide since none of scopes $\mathcal{S}_1, \mathcal{S}_2, \mathcal{S}_{1,2}$ is consistently redundant or suboptimal. Unsurprisingly, *rct* performs the worst among all algorithms.

**Experiment 2**  Consider the causal diagram in Fig. 4 where where $Y$ represents cardiovascular disease, $W$ blood pressure, $X_1$ taking an antihypertensive drug, and $X_2$ the use of an anti-diabetic drug [18]. Our goal is to evaluate policies with scopes $\mathcal{S}_1 = \{\langle X_1, \emptyset \rangle\}$, $\mathcal{S}_2 = \{\langle X_2, \emptyset \rangle\}$ and $\mathcal{S}_{1,2} = \{\langle X_1, \emptyset \rangle, \langle X_2, \{X_1\} \rangle\}$. We randomly generate 100 SCMs in Fig. 4 with binary $X_1, X_2, W, Y \in \{0, 1\}$. Exogenous variables $U_1, U_2, U_3$ are drawn from distributions over finite domains with cardinalities $d_1 = 8, d_2 = 12, d_3 = 16$ respectively. It follows from Thm. 2 such a parametric family is able to generate all expected rewards of policies with scopes $\mathcal{S}_1, \mathcal{S}_2, \mathcal{S}_{1,2}$. Simulation

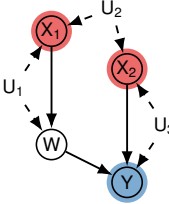

Figure 4

results, shown in Fig. 3b, reveal that the performance of *ts* and *ts\** coincide; the causal approach *c-ts\** consistently dominates *ts* and *ts\**; and finally, *rct* performs the worst among all strategies.

**Experiment 3**  Consider the causal diagram in Fig. 2 and policy scopes $\mathcal{S}_1 = \{\langle X_1, \{Z\} \rangle\}$, $\mathcal{S}_2 = \{\langle X_2, \{Z\} \rangle\}$ and $\mathcal{S}_{1,2} = \{\langle X_1, \{Z\} \rangle, \langle X_2, \{X_1, Z\} \rangle\}$. We randomly generate 100 SCMs in Fig. 2 with binary $X_1, X_2, Z, Y \in \{0, 1\}$. Exogenous variables $U_1, U_2$ are drawn from categorical distributions over domains with cardinalities $d_1 = 2, d_2 = 24$ respectively. Fig. 3c shows cumulative regrets averaging over random SCMs for all algorithms. Simulation results reveal that *c-ts\** consistently dominates other algorithms; *ts\** improves over *ts* since it does not explore the redundant policy scope $\mathcal{S}_{1,2}$. As expected, *rct* performs the worst among all learning strategies.

## 5    Conclusions

This paper investigated the online reinforcement learning for selecting an optimal treatment regimen from a policy space characterized with mixed state-action scopes. We first presented an online algorithm (Alg. 1) that achieves a sublinear regret but is computationally intractable for any moderated-size instance. We further introduced a novel type of parametrizations for general SCMs (Thm. 2), with finite observed and latent domains, that could represent an arbitrary subset of interventional distributions in a causal diagram using the minimal number of decomposing factors, called c-components. We then developed a more computationally efficient online algorithm (Alg. 3), based on the heuristics of Thompson sampling, that identifies an optimal policy with the same sample complexity. In today's healthcare, the growing use of combination therapies opens new opportunities in designing effective regimens by combining multiple treatments. The additional degrees of freedom present challenges in comparing different treatment regimens. We believe that our results constitute a significant step towards the development of a more principled science of personalized medicine.

## Acknowledgments and Disclosure of Funding

This research was supported in part by the NSF, ONR, DoE, Amazon, JP Morgan, and The Alfred P. Sloan Foundation.

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
