## A Proofs

In this section, we provide proofs for the theoretical results presented in the paper.

### A.1 Convergence of CAUSAL-UCB* and CAUSAL-TS*

Next we provide proofs for the regret bounds for CAUSAL-UCB* and CAUSAL-TS*. We begin by introducing some necessary lemmas. We first establish that the proposed $\hat{Q}[\boldsymbol{C}]$ is a consistent estimate for c-factor $Q[\boldsymbol{C}]$, for every $\boldsymbol{C} \in \mathbb{C}$. It will allow us to show that the confidence set $\mathscr{M}_t$ contains the underlying SCM $M$ with high probabilities.

**Lemma 1.** *For a causal diagram $\mathcal{G}$, let $\boldsymbol{C} \in \mathbb{C}$ be a c-component in $\mathcal{G}$. Then, c-factor $Q[\boldsymbol{C}]$ factorizes over a topological ordering $\prec$ in $\mathcal{G}$ as follows:*

$$Q[\boldsymbol{C}] = \prod_{V \in \boldsymbol{C}} q\left(v \mid \boldsymbol{pa}_V^+\right) \tag{6}$$

*where extended parents $\boldsymbol{PA}_V^+ = Pa(\boldsymbol{C}_V) \setminus \{V\}$; $\boldsymbol{C}_V$ is the c-component containing $V$ in $\mathcal{G}[\{V' \in \boldsymbol{C} \mid V' \prec V\}]$. Moreover, $q\left(V \mid \boldsymbol{PA}_V^+\right) = P\left(V \mid \boldsymbol{PA}_V^+, do(\pi)\right)$ for any policy $\pi \in \Pi_{\mathbb{S}(\boldsymbol{C})}$.*

*Proof.* The decomposition follows from the semi-Markovian factorization in [7, Def. 15]. □

**Lemma 2.** *Fix $\delta \in (0, 1)$. With probability (w.p.) $1 - \frac{\delta}{2}$, $M \in \mathscr{M}_t$ for all time steps $t = 1, 2, \ldots$.*

*Proof.* Fix a time step $t$. For every c-component $\boldsymbol{C} \in \mathbb{C}$ and every $V \in \boldsymbol{C}$ and any $\boldsymbol{pa}_V^+ \in \Omega_{\boldsymbol{PA}_V^+}$, define function $f_V(t, \delta)$ as

$$f_V(t, \delta) = \sqrt{\frac{6|\Omega_V| \ln\left(2|\Omega_{\boldsymbol{PA}_V^+}||\boldsymbol{V}(\mathbb{C})|t/\delta\right)}{\max\left\{n_t\left(\boldsymbol{pa}_V^+\right), 1\right\}}}. \tag{20}$$

Fix $n_t\left(\boldsymbol{pa}_V^+\right) = n$. It follows from the concentration inequality in [22, C.1] that

$$P\left(\left\|q\left(\cdot \mid \boldsymbol{pa}_V^+\right) - \hat{q}_t\left(\cdot \mid \boldsymbol{pa}_V^+\right)\right\|_1 > f_V(t, \delta) \text{ and } n_t\left(\boldsymbol{pa}_V^+\right) = n\right) \leq \frac{\delta}{4t^3|\Omega_{\boldsymbol{PA}_V^+}||\boldsymbol{V}(\mathbb{C})|} \tag{21}$$

Hence a union bound over all possible values of $n_t\left(\boldsymbol{pa}_V^+\right)$ implies that Eq. (4) holds at any time step $t$ with probability at most

$$P\left(\left\|q\left(\cdot \mid \boldsymbol{pa}_V^+\right) - \hat{q}_t\left(\cdot \mid \boldsymbol{pa}_V^+\right)\right\|_1 > f_V(t, \delta)\right) \leq \frac{\delta}{4t^2|\Omega_{\boldsymbol{PA}_V^+}||\boldsymbol{V}(\mathbb{C})|} \tag{22}$$

Summing these error probabilities over all realizations $\boldsymbol{pa}_V^+$ for every variable $V \in \boldsymbol{V}(\mathbb{C})$ gives $P\left(M \notin \mathscr{M}_t\right) \leq \frac{\delta}{4t^2}$. A union bound over all times steps $t = 1, 2, \ldots$ implies:

$$P\left(\forall t = 1, 2, \ldots, M \in \mathscr{M}_t\right) \geq 1 - \sum_{t=1}^{\infty} P\left(M \notin \mathscr{M}_t\right) \geq 1 - \sum_{t=1}^{\infty} \frac{\delta}{4t^2} \geq 1 - \frac{\delta}{2}. \tag{23}$$

This proves the claimed concentration bound. □

**Lemma 3.** *Fix $\delta \in (0, 1)$. W.p. at least $1 - \frac{\delta}{2}$, for any $T > 1$,*

$$\sum_{t=1}^{T} \mathbb{E}_M[Y \mid do(\pi_t)] - Y_t \leq \sqrt{2T \log(T/\delta)} \tag{24}$$

*Proof.* Let $Z_t = \mathbb{E}_M[Y \mid do(\pi_t)] - Y_t$ and let $\boldsymbol{H}_t = \{\boldsymbol{V}_i\}_{i=1}^{t-1}$ denote experimental history up to time step $t$. It is verifiable that $\mathbb{E}[Z_t \mid \boldsymbol{H}_t] = 0$ and $|Z_t| \leq 1$. This means that $Z_1, \ldots, Z_T$ is a sequence of martingale differences. Azuma-Hoeffding inequality [18] implies that for all $\epsilon > 0$ and $T \in \mathbb{N}$,

$$P\left(\sum_{t=1}^{T} Z_t > \epsilon\right) \leq \exp\left(-\frac{\epsilon^2}{2T}\right). \tag{25}$$

Setting $\epsilon = \sqrt{2T \log(T/\delta)}$ we obtain the claimed bound. □

**Lemma 4.** *Assume that $M \in \mathscr{M}_t$ for all time steps $t = 1, 2, \ldots$. Let $M_t$ be the solution of the inner maximization in Eq. (5). For all $\delta \in (0, 1)$ and $T > 1$,*

$$\sum_{t=1}^{T} \mathbb{E}_{M_t}[Y \mid do(\pi_t)] - \mathbb{E}_M[Y \mid do(\pi_t)] \leq 17\Delta(\mathcal{G}, \mathbb{S})\sqrt{|\mathbb{S}|T \ln\left(|\boldsymbol{V}(\mathbb{C})|T/\delta\right)}. \tag{26}$$

*Proof.* Let $\mathcal{S}_t$ denote the scope of policy $\pi_t$ at time step $t$. Let variables in $\boldsymbol{V}(\mathbb{C}_{\mathcal{S}_t})$ be ordered by $V^{(1)} \prec V^{(2)} \prec \cdots \prec V^{(k)}$ following a topological ordering in $\mathcal{G}_{\mathcal{S}_t}$. For any $i = 0, \ldots, k$, define

$$\mathbb{E}^{(i)}[Y \mid do(\pi_t)] = \sum_{\boldsymbol{v}(\mathbb{C}_{\mathcal{S}_t})\backslash y} y \prod_{j=1}^{i} P_M\left(v^{(j)} \mid \boldsymbol{pa}_{V^{(j)}}^+\right) \prod_{j=i+1}^{k} P_{M_t}\left(v^{(j)} \mid \boldsymbol{pa}_{V^{(j)}}^+\right). \tag{27}$$

By a telescoping sum, $\mathbb{E}_{M_t}[Y \mid do(\pi_t)] - \mathbb{E}_M[Y \mid do(\pi_t)]$ for any time $t$ could be written as

$$\mathbb{E}_{M_t}[Y \mid do(\pi_t)] - \mathbb{E}_M[Y \mid do(\pi_t)] = \sum_{i=0}^{k-1} \mathbb{E}^{(i)}[Y \mid do(\pi_t)] - \mathbb{E}^{(i+1)}[Y \mid do(\pi_t)]. \tag{28}$$

Observe that for any $i = 0, 1, \ldots, k$, expected rewards $\mathbb{E}^{(i)}[Y \mid do(\pi_t)]$ and $\mathbb{E}^{(i+1)}[Y \mid do(\pi_t)]$ only differ in the factor of $P\left(v^{(i)} \mid \boldsymbol{pa}_{V^{(i)}}^+\right)$. This implies

$$\mathbb{E}^{(i)}[Y \mid do(\pi_t)] - \mathbb{E}^{(i+1)}[Y \mid do(\pi_t)] \leq \sum_{\boldsymbol{pa}_{V^{(i)}}^+} \|P_{M_t}\left(\cdot \mid \boldsymbol{pa}_{V^{(i)}}^+\right) - P_M\left(\cdot \mid \boldsymbol{pa}_{V^{(i)}}^+\right)\|_1 \tag{29}$$

$$\leq \sum_{\boldsymbol{pa}_{V^{(i)}}^+} \|P_{M_t}\left(\cdot \mid \boldsymbol{pa}_{V^{(i)}}^+\right) - \hat{P}_t\left(\cdot \mid \boldsymbol{pa}_{V^{(i)}}^+\right)\|_1 \tag{30}$$

$$+ \sum_{\boldsymbol{pa}_{V^{(i)}}^+} \|\hat{P}_t\left(\cdot \mid \boldsymbol{pa}_{V^{(i)}}^+\right) - P_M\left(\cdot \mid \boldsymbol{pa}_{V^{(i)}}^+\right)\|_1 \tag{31}$$

Since both $M$ and $M_t$ is contained in the hypothesis class $\mathscr{M}_t$,

$$\mathbb{E}^{(i)}[Y \mid do(\pi_t)] - \mathbb{E}^{(i+1)}[Y \mid do(\pi_t)] \leq \sum_{\boldsymbol{pa}_{V^{(i)}}^+} 2\sqrt{\frac{6|\Omega_V| \ln\left(2|\Omega_{\boldsymbol{PA}_V^+}||\boldsymbol{V}(\mathbb{C})|t/\delta\right)}{\max\left\{n_t\left(\boldsymbol{pa}_V^+\right), 1\right\}}} \tag{32}$$

For any $\mathcal{S} \in \mathbb{S}$, let $T(\mathcal{S})$ be a subset of $\{1, \ldots, T\}$ containing time steps $t$ such that $\pi_t \sim \mathcal{S}$. We have

$$\sum_{t=1}^{T} \mathbb{E}_{M_t}[Y \mid do(\pi_t)] - \mathbb{E}_M[Y \mid do(\pi_t)] \tag{33}$$

$$= \sum_{\mathcal{S} \in \mathbb{S}} \sum_{t \in T(\mathcal{S})} \mathbb{E}_{M_t}[Y \mid do(\pi_t)] - \mathbb{E}_M[Y \mid do(\pi_t)] \tag{34}$$

$$= \sum_{\mathcal{S} \in \mathbb{S}} \sum_{t \in T(\mathcal{S})} \sum_{i=0}^{k-1} \mathbb{E}^{(i)}[Y \mid do(\pi_t)] - \mathbb{E}^{(i+1)}[Y \mid do(\pi_t)] \tag{35}$$

$$\leq \sum_{\mathcal{S} \in \mathbb{S}} \sum_{t \in T(\mathcal{S})} \sum_{i=0}^{k-1} 2\sqrt{\frac{6|\Omega_V| \ln\left(2|\Omega_{\boldsymbol{PA}_V^+}||\boldsymbol{V}(\mathbb{C})|t/\delta\right)}{\max\left\{n_t\left(\boldsymbol{pa}_V^+\right), 1\right\}}} \tag{36}$$

Let $n_t(\mathcal{S})$ denote the total occurrence of event $\pi_t \sim \mathcal{S}$ prior to time $t$. Applying [22, C.3] gives

$$\sum_{t=1}^{T} \mathbb{E}_{M_t}[Y \mid do(\pi_t)] - \mathbb{E}_M[Y \mid do(\pi_t)] \tag{37}$$

$$\leq \sum_{\mathcal{S} \in \mathbb{S}} \sum_{V \in \boldsymbol{V}(\mathbb{C}_{\mathcal{S}})} 12\sqrt{|\Omega_{V \cup \boldsymbol{PA}_V^+}|n_{T+1}(\mathcal{S}) \ln\left(2|\Omega_{\boldsymbol{PA}_V^+}||\boldsymbol{V}(\mathbb{C})|T/\delta\right)} \tag{38}$$

$$\leq \sum_{\mathcal{S} \in \mathbb{S}} \sqrt{n_{T+1}(\mathcal{S})} \max_{\mathcal{S} \in \mathbb{S}} \sum_{V \in \boldsymbol{V}(\mathbb{C}_{\mathcal{S}})} 12\sqrt{|\Omega_{V \cup \boldsymbol{PA}_V^+}| \ln\left(2|\Omega_{\boldsymbol{PA}_V^+}||\boldsymbol{V}(\mathbb{C})|T/\delta\right)} \tag{39}$$

Applying Jensen's inequality we obtain

$$\sum_{t=1}^{T} \mathbb{E}_{M_t}[Y \mid \mathrm{do}(\pi_t)] - \mathbb{E}_M[Y \mid \mathrm{do}(\pi_t)] \tag{40}$$

$$\leq \max_{S \in \mathbb{S}} \sum_{V \in \boldsymbol{V}(\mathbb{C}_S)} 12 \sqrt{|\Omega_{V \cup \boldsymbol{PA}_V^+}| |\mathbb{S}| T \ln \left(2|\Omega_{\boldsymbol{PA}_V^+}| |\boldsymbol{V}(\mathbb{C})| T / \delta\right)}$$

A few simplification gives the claimed bound

$$\sum_{t=1}^{T} \mathbb{E}_{M_t}[Y \mid \mathrm{do}(\pi_t)] - \mathbb{E}_M[Y \mid \mathrm{do}(\pi_t)] \leq 17 \Delta(\mathcal{G}, \mathbb{S}) \sqrt{|\mathbb{S}| T \ln \left(|\boldsymbol{V}(\mathbb{C})| T / \delta\right)} \tag{41}$$

where function $\Delta(\mathcal{G}, \mathbb{S}) = \max_{S \in \mathbb{S}} \sum_{V \in \boldsymbol{V}(\mathbb{C}_S)} \sqrt{|\Omega_{V \cup \boldsymbol{PA}_V^+}|}$. □

**Theorem 1.** *For a causal diagram $\mathcal{G}$ and a mixed policy scope $\mathbb{S}$, fix a $\delta \in (0, 1)$. With probability at least $1 - \delta$, it holds for any $T > 1$, the regret of* CAUSAL-UCB\* *is bounded by*

$$R(T, M) \leq 19 \Delta(\mathcal{G}, \mathbb{S}) \sqrt{|\mathbb{S}| T \ln \left(|\boldsymbol{V}(\mathbb{C})| T / \delta\right)}. \tag{9}$$

*where function $\Delta(\mathcal{G}, \mathbb{S}) = \max_{S \in \mathbb{S}} \Delta(\mathcal{G}, \mathcal{S})$ and $\Delta(\mathcal{G}, \mathcal{S}) = \sum_{V \in \boldsymbol{V}(\mathbb{C}_S)} \sqrt{|\Omega_{V \cup \boldsymbol{PA}_V^+}|}$.*

*Proof.* The cumulative regret $R(T, M)$ could be written as follows, by a telescoping sum:

$$R(T, M) = \sum_{t=1}^{T} \mathbb{E}_M[Y \mid \mathrm{do}(\pi^*)] - Y_t \tag{42}$$

$$= \sum_{t=1}^{T} \mathbb{E}_M[Y \mid \mathrm{do}(\pi^*)] - \mathbb{E}_M[Y \mid \mathrm{do}(\pi_t)] + \sum_{t=1}^{T} \mathbb{E}_M[Y \mid \mathrm{do}(\pi_t)] - Y_t \tag{43}$$

Lem. 2 implies that w.p. $1 - \frac{\delta}{2}$, the actual SCM $M \in \mathscr{M}_t$ for all time steps $t$. Since $M_t$ and $\pi_t$ are the optimistic instance in $\mathscr{M}_t$ that achieves the maximal expected reward,

$$R(T, M) \leq \sum_{t=1}^{T} \mathbb{E}_{M_t}[Y \mid \mathrm{do}(\pi_t)] - \mathbb{E}_M[Y \mid \mathrm{do}(\pi_t)] + \sum_{t=1}^{T} \mathbb{E}_M[Y \mid \mathrm{do}(\pi_t)] - Y_t \tag{44}$$

Among quantities in the above equation, Lem. 3 implies that w.p. $1 - \frac{\delta}{2}$,

$$\sum_{t=1}^{T} \mathbb{E}_M[Y \mid \mathrm{do}(\pi_t)] - Y_t \leq \sqrt{2T \log(T/\delta)} \tag{45}$$

Applying Lem. 4 gives the following bound:

$$\sum_{t=1}^{T} \mathbb{E}_{M_t}[Y \mid \mathrm{do}(\pi_t)] - \mathbb{E}_M[Y \mid \mathrm{do}(\pi_t)] \leq 17 \Delta(\mathcal{G}, \mathbb{S}) \sqrt{|\mathbb{S}| T \ln \left(|\boldsymbol{V}(\mathbb{C})| T / \delta\right)}. \tag{46}$$

The above equations together imply

$$R(T, M) \leq 17 \Delta(\mathcal{G}, \mathbb{S}) \sqrt{|\mathbb{S}| T \ln \left(|\boldsymbol{V}(\mathbb{C})| T / \delta\right)} + \sqrt{2T \log(T/\delta)} \tag{47}$$

$$\leq 19 \Delta(\mathcal{G}, \mathbb{S}) \sqrt{|\mathbb{S}| T \ln \left(|\boldsymbol{V}(\mathbb{C})| T / \delta\right)} \tag{48}$$

The error probabilities are bounded by $\frac{\delta}{2} + \frac{\delta}{2} = \delta$. This proves the claimed regret bound. □

**Theorem 3.** *Given a causal diagram $\mathcal{G}$, a mixed policy scope $\mathbb{S}$, and a prior distribution $\rho$, it holds for any $T > 1$, the regret of* CAUSAL-TS\* *is bounded by*

$$R(T, \rho) \leq 26 \Delta(\mathcal{G}, \mathbb{S}) \sqrt{|\mathbb{S}| T \ln \left(|\boldsymbol{V}(\mathbb{C})| T\right)}. \tag{19}$$

*Proof.* The idea of the proof was established in [42, 34]. First, note that given any sample history $\boldsymbol{H}_t = \{\boldsymbol{V}_i\}_{i=1}^{t-1}$, the true SCM $M$ and the sampled $M_t$ are identically distributed. That means that

$$\mathbb{E}\left[\mathbb{E}_M[Y \mid \mathrm{do}(\pi^*)] \mid \boldsymbol{H}_t, M \sim \rho\right] = \mathbb{E}\left[\mathbb{E}_{M_t}[Y \mid \mathrm{do}(\pi_t)] \mid \boldsymbol{H}_t, M_t \sim \rho\right] \tag{49}$$

Since $Y_t \sim P(Y \mid \mathrm{do}(\pi_t))$, we also have

$$\mathbb{E}\left[Y_t \mid \boldsymbol{H}_t, M \sim \rho\right] = \mathbb{E}\left[\mathbb{E}_M[Y \mid \mathrm{do}(\pi_t)] \mid \boldsymbol{H}_t, M \sim \rho\right] \tag{50}$$

The Bayesian cumulative regret $R(T, \rho)$ could thus be written as:

$$R(T, \rho) = \mathbb{E}\left[\sum_{t=1}^{T} \mathbb{E}_M[Y \mid \mathrm{do}(\pi^*)] - Y_t \mid M \sim \rho\right] \tag{51}$$

$$= \mathbb{E}\left[\sum_{t=1}^{T} \mathbb{E}_{M_t}[Y \mid \mathrm{do}(\pi_t)] - Y_t \mid M, M_t \sim \rho\right] \tag{52}$$

$$= \mathbb{E}\left[\sum_{t=1}^{T} \mathbb{E}_{M_t}[Y \mid \mathrm{do}(\pi_t)] - \mathbb{E}_M[Y \mid \mathrm{do}(\pi_t)] \mid M, M_t \sim \rho\right] \tag{53}$$

We can use $\rho(M \mid \boldsymbol{H}_t) = \rho(M_t \mid \boldsymbol{H}_t)$ again and say that both $M$ and $M_t$ belongs to $\mathscr{M}_t$ for all time steps $t$ w.p. at least $1 - \delta$ (Lem. 2). This means that we can bound $R(T, \rho)$ in CAUSAL-TS*

$$R(T, \rho) \leq \mathbb{E}\left[\sum_{t=1}^{T} \mathbb{E}_{M_t}[Y \mid \mathrm{do}(\pi_t)] - \mathbb{E}_M[Y \mid \mathrm{do}(\pi_t)] \mid M, M_t \in \mathscr{M}_t\right] + \delta T \tag{54}$$

$$\leq 17\Delta(\mathcal{G}, \mathbb{S})\sqrt{|\mathbb{S}|T \ln\left(|\boldsymbol{V}(\mathbb{C})|T/\delta\right)} + \delta T \tag{55}$$

The last step follows from Lem. 4. Setting $\delta = \frac{1}{T}$ we obtain the claimed bound. $\qquad\square$

## A.2 $\mathbb{C}$-Canonical SCMs

In this section, we provide proofs for theoretical results related to $\mathbb{C}$-canonical SCMs.

**Theorem 2.** *For any SCM $M = \langle \boldsymbol{V}, \boldsymbol{U}, \mathscr{F}, P(\boldsymbol{U})\rangle$, let $\mathbb{C}$ be an arbitrary c-collection. For any $\boldsymbol{C} \in \mathbb{C}$, c-factor $Q[\boldsymbol{C}]$ decomposes as follows:*

$$Q[\boldsymbol{C}](\boldsymbol{v}) = \sum_{U \in \boldsymbol{U}} \sum_{u=1,\ldots,d_U} \prod_{V \in \boldsymbol{C}} \mathbb{1}\{f_V(\boldsymbol{pa}_V, \boldsymbol{u}_V) = v\} \prod_{U \in \boldsymbol{U}} P(u) \tag{11}$$

*where for every exogenous $U \in \boldsymbol{U}$, $P(U)$ is a discrete distribution over a finite domain $\{1, \ldots, d_U\}$ with cardinality $d_U = \sum_{\boldsymbol{C} \in \mathbb{C}(U)} |\Omega_{Pa(\boldsymbol{C})}|$; $\mathbb{C}(U) \subseteq \mathbb{C}$ are c-components covering $U$.*

*Proof.* Let $\vec{P}$ be a vector representing all values of c-factors $Q[\boldsymbol{C}]$ contained in $\mathbb{C}$. Formally,

$$\vec{P} = (Q[\boldsymbol{C}](\boldsymbol{c}, \boldsymbol{pa_C}) \mid \forall \boldsymbol{C} \in \mathbb{C}, \forall \boldsymbol{c} \in \Omega_{\boldsymbol{C}}, \forall \boldsymbol{pa_C} \in \Omega_{\boldsymbol{PA_C}}) \tag{56}$$

where $\boldsymbol{PA_C} = Pa(\boldsymbol{C}) \setminus \boldsymbol{C}$. Obviously, $\vec{P}$ is a vector containing $d = \sum_{\boldsymbol{C} \in \mathbb{C}(U)} |\Omega_{Pa(\boldsymbol{C})}|$ elements. However, since for any $\boldsymbol{C} \in \mathbb{C}$, $\sum_{\boldsymbol{c}} Q[\boldsymbol{C}] = 1$, it only takes a vector with $d - 1$ dimensions to determine $\vec{P}$. We could thus see $\vec{P}$ as a point in the $(d-1)$-dimensional real space. Following the discretization procedure in [60, Lem. A.6] we obtain the claimed decomposition. $\qquad\square$

**Proposition 1.** *For a causal diagram $\mathcal{G}$ and a c-collection $\mathbb{C}$, MINCOLLECT$(\mathcal{G}, \mathbb{C})$ returns a minimal reduction $\mathbb{C}^*$ of $\mathbb{C}$.*

*Proof.* The soundness of IDENTIFY implies that MINCOLLECT must returns a valid reduction $\mathbb{C}^*$ of c-collection $\mathbb{C}$ in $\mathcal{G}$. What remains is to show that $\mathbb{C}^*$ is minimal. Suppose $\mathbb{C}^*$ is not minimal. That is, there exists a c-component $\boldsymbol{C} \in \mathbb{C}^*$ such that $Q[\boldsymbol{C}]$ is identifiable from other c-factors $Q[\boldsymbol{C}']$ in $\mathbb{C}^* \setminus \{\boldsymbol{C}\}$ in $\mathcal{G}$. It follows from the construction procedure in [31, Theorem 3] that one could construct a pair of SCMs $M_1, M_2$ compatible with $\mathcal{G}$ such that $Q_{M_1}[\boldsymbol{C}] \neq Q_{M_2}[\boldsymbol{C}]$ while $Q_{M_1}[\boldsymbol{C}'] = Q_{M_2}[\boldsymbol{C}']$ for any other $\boldsymbol{C}' \in \mathbb{C}^* \setminus \{\boldsymbol{C}\}$. This means that $Q[\boldsymbol{C}]$ is not identifiable from $\mathbb{C}^* \setminus \{\boldsymbol{C}\}$ in $\mathcal{G}$, which is a contradiction. $\qquad\square$

**Proposition 2.** *For a causal diagram $\mathcal{G}$, any c-collection $\mathbb{C}$ has a unique minimal reduction.*

*Proof.* We will utilize the following claim

**Claim 1.** *If $\mathbb{C}_1$ and $\mathbb{C}_2$ are both reductions of a c-collection $\mathbb{C}$ in a causal diagram $\mathcal{G}$, then $\mathbb{C}_1 \cap \mathbb{C}_2$ is a reduction of both $\mathbb{C}_1$ and $\mathbb{C}_2$ in $\mathcal{G}$.*

The uniqueness of the minimal reduction of any c-collection $\mathbb{C}$ follows immediately from the above claim. Suppose $\mathbb{C}$ has two different minimal reductions $\mathbb{C}_1, \mathbb{C}_2$. Claim 1 implies that their intersection $\mathbb{C}_1 \cap \mathbb{C}_2$ is a reduction of $\mathbb{C}_1$ and $\mathbb{C}_2$, which contradicts the assumption that $\mathbb{C}_1$ and $\mathbb{C}_2$ are both minimal reductions.

Next we will provide the proof for Claim 1. Let $m_i = |\mathbb{C} \setminus \mathbb{C}_i|$ where $i = 1, 2$. We will show the result by induction after $m = m_1 + m_2$.

**Base Case:** $m = 2$. Let $\boldsymbol{C}_i = \mathbb{C} \setminus \mathbb{C}_i$ for $i = 1, 2$. It suffices to show that $\boldsymbol{C}_1 \in \mathbb{C}_2$ is identifiable from c-factors in $\mathbb{C}_2 \setminus \{\boldsymbol{C}_1\}$. MINCOLLECT shows that $\boldsymbol{C}_1$ is identifiable from $\mathbb{C}_1$ if any only if there exists a c-component $\boldsymbol{C} \in \mathbb{C}_1$ such that $Q[\boldsymbol{C}_1]$ is identifiable from $Q[\boldsymbol{C}]$. If $\boldsymbol{C} \neq \boldsymbol{C}_2$, we must have $\mathbb{C}_1 \cap \mathbb{C}_2 = \mathbb{C}_2 \setminus \{\boldsymbol{C}_1\}$ is a reduction of $\mathbb{C}_2$ since $\boldsymbol{C} \in \mathbb{C}_2$ and IDENTIFY$(\boldsymbol{C}_1, \boldsymbol{C}, \mathcal{G}) \neq$ FAIL. If $\boldsymbol{C} = \boldsymbol{C}_2$, since $\mathbb{C}_2$ is a reduction of $\mathbb{C}$ by removing $\boldsymbol{C}_2$, there must exists c-component $\boldsymbol{C}' \in \mathbb{C}$ such that IDENTIFY$(\boldsymbol{C}_2, \boldsymbol{C}', \mathcal{G}) \neq$ FAIL. Also, note that $\boldsymbol{C}' \neq \boldsymbol{C}_1$; otherwise, one would have $\boldsymbol{C}_1 = \boldsymbol{C}_2$ which contradicts the fact that $\mathbb{C}_1 \neq \mathbb{C}_2$. This means that $\boldsymbol{C}' \in \mathbb{C}_1 \cap \mathbb{C}_2$, which again implies that $\boldsymbol{C}_1$ is identifiable from $\mathbb{C}_2 \setminus \{\boldsymbol{C}_1\}$ in $\mathcal{G}$. We could thus obtain a reduction $\mathbb{C}_1 \cap \mathbb{C}_2$ of $\mathbb{C}_2$ by removing c-component $\boldsymbol{C}_1$. The proof for $\mathbb{C}_1 \cap \mathbb{C}_2$ being a reduction of $\mathbb{C}_1$ follows the same procedure.

**Induction Step:** $m \leq k + 1$. Suppose the result holds for $m \leq k$ where $k \geq 2$ and consider the case $m = k + 1$. So $\max\{m_1, m_2\} > 1$, say $m_2 > 1$. Thus $\mathbb{C}_2$ is obtained by successively removing $m_2$ identifiable c-components from $\mathbb{C}$. Let $\mathbb{C}'_2$ be a reduction obtained by removing the first $m_2 - 1$ of these. By the induction assumption, $\mathbb{C}_1 \cap \mathbb{C}'_2$ is a reduction of $\mathbb{C}_2$ obtained by removing at most $m_1$ identifiable c-components from $\mathbb{C}'_2$. Furthermore, $\mathbb{C}_2$ is also a reduction of $\mathbb{C}'_2$ obtained by removing exactly one identifiable c-component. Since $(\mathbb{C}_1 \cap \mathbb{C}'_2) \cap \mathbb{C}_2 = \mathbb{C}_1 \cap \mathbb{C}_2$ and $m_1 + 1 \leq k$, the induction assumption yields that $\mathbb{C}_1 \cap \mathbb{C}_2$ is a reduction of $\mathbb{C}_2$. Similarly, the induction assumption gives that $\mathbb{C}_1 \cap \mathbb{C}_2$ is a reduction of $\mathbb{C}_1$. This completes the proof. $\qquad\square$

# B  Simulation Setups

In all experiments, we evaluate our proposed CAUSAL-TS* with uninformative Dirichlet priors over exogenous probabilities and uniform priors over structural functions, which we label as *c-ts\**. As a baseline, we also include following algorithms. (1) Randomized trials (*rct*) allocating treatments in all possible scopes uniformly at random; (2) standard Thompson sampling algorithm (*ts*) using all deterministic policies as arms; and (3) Thompson sampling over a simplified mixed scope (*ts\**), which is obtained by applying graphical conditions in [30]. For each experiment, we randomly generate 100 instances of SCMs compatible with the corresponding causal diagram. For each random SCM, we measure the cumulative regrets of all algorithms over $T = 1.1 \times 10^3$ episodes. For every algorithm in every random SCM instance, we repeat the online learning process for 100 times, and compute the cumulative regret averaging over all repetitions. Finally, all experiments were performed on a computer with 32GB memory, implemented in MATLAB.

# C  Related Work on Canonical SCMs

The idea of canonical SCMs was first explored in [5, 4], which introduced a canonical partitioning of exogenous domains in the 'IV" diagram in Fig. 5. For binary endogenous variables $X, Y, Z \in \{0, 1\}$, the canonical partitioning allows one to discretize the domain of $U$ into 16 equivalent classes without changing the original counterfactual distributions and the graphical structure in Fig. 5. Such discritization is also referred to as the principal stratification [15, 37]. Based on this finite-state representation, tight bounding strategy was proposed to evaluate treatment effects under the condition of imperfect compliance in randomized experiments [6]. There

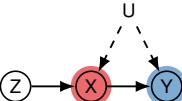

Figure 5: IV

also exist Bayesian approaches to obtain posterior distributions of causal effects provided with data collected from experimental studies with imperfect compliance [12, 20].

The canonical partitioning could be extended to a more generalized class of causal diagrams that are reducible to the "IV" graph [43, 56]. However, these methods do not necessarily encode all constraints over induced distributions. [14] showed that for a specific class of causal diagrams satisfying a running intersection property among exogenous variables, all equality and inequality constraints over the observational distribution could be generated using discrete unobserved domains. [41] applied a classic result of Carathéodory theorem in convex geometry [9] and developed a generative model with finite-state unobserved variables that could represent the observational distribution over discrete domains in an arbitrary causal diagram. More recently, [60] introduced a family of canonical SCMs that could represent all categorical counterfactual distributions in any causal diagram with finite exogenous states. Using this canonical representation, the problem of inferring counterfactual probabilities from the combination of observational and interventional data is reducible to a series of polynomial optimization programs [55, 13]. The computational framework of neural networks is also applicable to determine the identifiability of causal effects over discrete observed domains [54]. Finally, representing distributions over continuous observed domains is more challenging; existing methods often require untestable parametric assumptions about the underlying environment [23, 19].

Thm. 2 extends existing results in several non-trivial ways. First, the cardinality of exogenous states in canonical SCMs [60] grows exponentially with regard to the total number of observed states. The restricted family of canonical SCM in Thm. 2 is tailored for an arbitrary collection of c-components in the causal diagram. This means that, in many cases, the cardinality of exogenous domains in $\mathbb{C}$-canonical SCMs could be sparse, growing as a polynomial function of the size of observed states. Second, we propose a novel algorithm (Alg. 2) to exploit equality relationships among c-factors. This allows us to further reduce the model complexity of $\mathbb{C}$-canonical SCMs while maintaining the same qualitative and quantitative constraints over parameters of target c-factors.