# OpenReview forum: "Online Reinforcement Learning for Mixed Policy Scopes"
_NeurIPS.cc/2022/Conference — NeurIPS 2022 Accept_

### Official Review · Reviewer_CZeJ · 2022-07-07

**Rating:** 7
**Confidence:** 2
**Soundness:** 4 excellent
**Presentation:** 4 excellent
**Contribution:** 3 good

**Summary:**

This paper considers online policy learning with different policy scopes in the context of causal inference. The paper develops online learning algorithms with sub-linear regrets and presents simulation results of the theoretical algorithms.

**Questions:**

My only question is regarding the statement that most of the works are based on the offline learning setting. However, I do have an impression that there are many causal learning algorithms (even on the practical RL side) that are online. Is the major difference here primarily about the policy scopes?

**Strengths And Weaknesses:**

First of all, I have to be frank that I'm only knowledgeable of causal inference as a practical RL person. So most of my comments are from a rough understanding of this draft.

I feel the paper is clearly written with rigorous (possibly textbook-style) formulations and derivations. The motivation is clear, and all the proofs and introduced algorithms look correct. The simulation results also look reasonable as a theoretical paper. Although I cannot confidently judge the novelty of this work, I would still rate it positive for this work.

---

> ### Author Response · Authors · 2022-08-02
> **Response to Reviewer CZeJ**
>
> We thank the reviewer for the positive feedback and assessment of our work. We have addressed below the reviewer’s question regarding the novelty of the work.
>
> ---
> >#### Q1. "My only question is regarding the statement that most of the works are based on the offline learning setting. However, I do have an impression that there are many causal learning algorithms (even on the practical RL side) that are online. Is the major difference here primarily about the policy scopes?"
>
> Many online learning algorithms indeed exist for optimizing candidate policies in an unknown structural causal model (SCM), including C-UCB (Lu, Yangyi, et al. 2020) and, more generally, OFU-DTR (Zhang & Bareinboim, 2020). However, these methods assume all candidate policies share the same state-action scope. Therefore, they are not applicable for the more generalized setting where the policies scopes are mixed, containing different combinations of action variables. A good analogy in RL literature is the standard bandits and combinatorial bandits. UCB algorithms designed for bandits might not achieve reasonable performance in the combinatorial bandit setting, thus calling for a separate investigation. More recently, (29, Lee & Bareinboim 2020) studied the graphical characterization of the combinatorial space of dynamic treatment regimes, i.e., the mixed policy scope. However, their approach focuses on the offline setting and does not always lead to the identification of optimal policies. This paper explores a new dimension of this problem setting and studies online learning over the combinatorial space of dynamic treatment regimes.
>
> One of the main challenges in designing an effective online algorithm is to compute the upper confidence bounds using data collected from policies with different state-action scopes. That is, one might be able to utilize the “information leakage” in data collected by policies with different scopes to accelerate the learning process. We address this challenge by introducing a novel parametric family of canonical SCMs (Theorem 2) that could represent the expected rewards of all candidate policies in a causal diagram. The construction of this canonical SCM is non-trivial, depending on both the network structure of the causal diagram and the target mixed scope. We developed a novel procedure (MinCollect) to exploit functional relationships among interventional distributions to reduce the cardinality of exogenous (latent) variables. We believe that this canonical family of SCMs could also be useful for the offline learning setting, especially when the expected rewards are not uniquely identifiable from data due to the presence of unobserved confounding. Possible applications include partial identification (Kallus & Zhou, 2018, 2020; Namkoong, Keramati, Yadlowsky & Brunskill 2020), where the agent attempts to obtain bounds over the expected rewards from biased/observational data.

---

### Official Review · Reviewer_NrX5 · 2022-07-10

**Rating:** 6
**Confidence:** 2
**Soundness:** 3 good
**Presentation:** 2 fair
**Contribution:** 3 good

**Summary:**

This paper considers the problem of effective *online* learning of optimal policies with mixed scopes (e.g. different types of treatments) in an arbitrary causal system.
The authors propose an algorithm (CAUSAL-UCB) to solve the problem and prove that it achieves sublinear regret.
Furthermore, to improve computational efficiency, the authors introduce a canonical representation for structural casual models and consider minimal reduction of c-components, then propose CAUSAL-TS as an easy-to-implement alternative.
The TS algorithm is evaluated on simple causal graphs to demonstrate its advantage over simple baselines.

**Questions:**

- For Theorem 3, I don't think we can convert UCB regret to TS regret for arbitrary problem instances. Please include exact conditions in the Theorem statement and rigorous proof.

- Is it possible to extend the current algorithm to endogenous variables with possibly infinite support? What would be the difficult part? Or, if this is too much, it is good to at least provide experimental results where the endogenous variables take more than two values.

- Please include a summary of notations in the appendix. It would be great if more intuitive explanations in words can be provided (e.g. what do they mean / why do they matter).

**Limitations:**

All my concerns are listed in the previous sessions. Please respond to questions and suggestions raised in "Questions".


**Strengths And Weaknesses:**

**Strengths**
- Originality
    - This paper introduces the novel problem of effective *online* learning of optimal policies with mixed scopes in an arbitrary causal system. The authors provide a UCB-based algorithm that solves this problem, as well as new ideas to improve computational efficiency.
    - Relevant works are cited, and the authors make the contribution clear.
- Clarity
    - The paper is overall well-structured, and I can get the main points quickly.
    - The paper is heavy in notations, but the authors make use of simple examples to explain them.
- Significance
    - I think the problem considered in this paper is very important and relevant in practice, and the current paper can motivate more research in online learning and causal inference.

**Weaknesses**
- Quality
    - Certain claims (e.g. Theorem 3) do not seem well-supported. Seem Questions session for details.
    - The instances in the experiments are somewhat over-simple.
- Clarity
    - This paper is VERY heavy in notations, and I find it hard to memorize and digest some of these notations when I first read the paper. A summary table is highly recommended.
- Significance
    - The algorithm heavily depends on the assumption that endogenous variables have finite supports. But in many practical settings, many endogenous variables are not discreet (especially for contexts), so I'm a bit doubtful about the practicality of the proposed methods.

---

> ### Author Response · Authors · 2022-08-02
> **Response to Reviewer NrX5 [1/2]**
>
> We thank the reviewer for the thoughtful and constructive comments, and we tried our best to address them accordingly.
>
> ---
> >#### Q1. "For Theorem 3, I don't think we can convert UCB regret to TS regret for arbitrary problem instances. Please include exact conditions in the Theorem statement and rigorous proof."
>
> Theorem 3 focuses on the analysis of Bayesian regret (41, Russo & Van Roy, 2014), which is the cumulative regret averaging over all possible SCMs drawn from the prior distribution $\rho$. A detailed and rigorous proof is provided in Appendix A, but we would be happy to improve in case you find something unclear. In general, the regret bound in Theorem 3 holds when the underlying SCM $M$ follows the prior $\rho$. In this paper, we assume that exogenous probabilities $P(U)$ are drawn from non-informative Dirichlet distributions (Eq. 17) since Theorem 2 ensures that one could assume the latent domain to be discrete and finite without loss of generality. Furthermore, the regret bound of Theorem 1 can be further extended to more generalized settings in which Bayesian regret is robust to prior misspecification, following the argument in (41, Russo & Van Roy, 2014).
>
> ---
> >#### Q2. "Is it possible to extend the current algorithm to endogenous variables with possibly infinite support? What would be the difficult part? Or, if this is too much, it is good to at least provide experimental results where the endogenous variables take more than two values."
>
> A critical component in designing online algorithms for the mixed policy scope is a novel parametric family of canonical SCMs (Theorem 2) that could represent the expected rewards of all candidate policies in a causal diagram over finite observed domains. Still, significant challenges exist in designing canonical SCMs that could represent all interventional distributions in an arbitrary causal diagram over infinite support. Also, the regret analysis using this type of canonical SCMs over infinite observed domains requires some new intricate concentration inequalities. After all, online reinforcement learning in unknown structural causal models with infinite support is an exciting challenge, which we plan to explore further.
>
> As for the second part of the question, we perform additional simulations for Figure 2 where all variables take values in 5 possible outcomes. We evaluate the novel Causal-TS*, with uninformative Dirichlet priors over exogenous probabilities and uniform priors over structural functions, which we label as c-ts*. As a baseline, we also include randomized trials (rc}) allocating treatments in all possible scopes uniformly at random, standard Thompson sampling algorithm (ts) using all deterministic policies as arms, and Thompson sampling over a simplified mixed scope (ts*), Simulation results measuring cumulative regrets over $T = 5000$ episodes are provided below. One could see by inspection that the causal approach c-ts* consistently dominates other algorithms. We will include these additional simulations in the updated manuscript.
>
> |      | rct | ts | ts* | c-ts* |
> |--------|---------|---------|---------|---------|
> | Cumulative Regret | $5951$ | $5977$ | $5997$ | $2931$ |

---

> ### Author Response · Authors · 2022-08-02
> **Response to Reviewer NrX5 [2/2]**
>
> ---
> >#### Q3. "Please include a summary of notations in the appendix. It would be great if more intuitive explanations in words can be provided (e.g. what do they mean / why do they matter)."
>
> Thank you for the suggestion. We provide below a table describing some of the necessary notations. Also, we will include a summary of notations in the updated manuscript.
>
> |   Notation     | Description |
> |--------|---------|
> | $M$ | Structural Causal Model(SCM) |
> | $\mathcal{G}$ | Causal diagram associated with the SCM  |
> | $P(U)$ | Exogenous distribution over unobserved variables $U$  |
> | $P(V)$ | Observational distribution over endogenous (observed) variables $V$  |
> | $\Omega_V$ | Domain of a variable $V$ |
> | $X$ | Action variables that can be intervened on |
> | $S_X$ | Observed covariates associated with an action $X$ |
> | $\mathcal{S}$ | A policy scope describing actions $X$ that an agent could intervene on and associated covariates $S_X$|
> | $\pi$ | A policy compatible with a policy scope, consisting of conditional distribution $\pi(X \mid S_X)$ |
> | $\mathbb{S}$ | A mixed scope containing a collection of policy scopes |
> | $\Pi_{\mathbb{S}}$ | Candidate policies compatible with a mixed scope |
> | $P(Y \mid do(\pi))$ | Interventional distribution induced by a policy $\pi$ |
> | $P(Y \mid do(x))$ | Interventional distribution induced by a atomic policy $\pi: X \gets x$ which sets values of actions $X$ to a constant|
> | $Q[C]$ | C-factor over variables $C$, equal to interventional distributions $P(c \mid do(v \backslash c))$|
> | $C$ | C-component consisting of observed variables connected by bi-directed paths|
> | $\mathbb{C}$ | A collection of c-components |
> | $\mathbb{C}(\mathcal{G})$ | C-components contained in a causal diagram |
> | $\mathbb{C}_{\mathbb{S}}$ | C-components contained in a causal diagram induced by policies with a mixed scope $\mathbb{S}$ |
> | $\mathbb{S}(C)$ | A mixed scope that could generate a c-component |

---

### Official Review · Reviewer_NFf7 · 2022-07-10

**Rating:** 5
**Confidence:** 3
**Soundness:** 3 good
**Presentation:** 3 good
**Contribution:** 2 fair

**Summary:**

The paper provides an algorithm for online learning in causal systems with mixed policy scopes that has sub-linear regret.  Results are presented on (small) simulated domains.

**Questions:**

- Can you provide more detailed analysis of the space/time analysis for the algorithms CAUSAL-UCB* and the MINCOLLECTs?    The simulation results suggest this is limited to very small number of state/action variables; is this due to learning efficiency (in terms of #samples) or computational efficiency (in terms of memory/#operations)?  Could you include simulation results as a function of different factors (delta, #states/actions)?
- Considering Experiment 2, only blood pressure is considered as a variable (not even age, gender and other well-known contributing factors).  Is this because the method is limited to such small domain?  Or because it necessitates strong assumptions?


**Limitations:**

- The paper does not really discuss technical limitations of the work, except where it serves to motivate the proposed approach.
- There could be a deeper discussion of the ethical aspects of this work, given the proposed application to selecting optimal treatment regime, and the example on patients with cardiovascular disease.


**Strengths And Weaknesses:**

(+)	The paper is motivated by real-world practical problem in treatment optimization.

(+)	The paper is well grounded in terms of theoretical framework and includes results on sub-linear learning rate for the proposed setting.

(+)	The results of sub-linear regret appear sound theoretically and are confirmed by simulations.

(-)	I wonder how useful & interesting is the proposed setting, grounded in causal learning, compared to more standard MDP settings.  I’m less familiar with this literature (but very familiar with MDP/RL literature), and it made it quite challenging to follow precisely the setting, algorithm and theoretical analysis.

(-)	The proposed method appears limited to very small settings (few variables). I do not find the results very convincing given their simplicity.  See specific questions below about Experiment 2.

(-)	The empirical results show an improvement for the proposed c-ts* method, compared to Thompson sampling alternatives. However the gap seems relatively small and both approaches seem sub-linear.  How do you know if the improvement of c-ts* is significant?  Is the results really meaningful?

---

> ### Author Response · Authors · 2022-08-02
> **Response to Reviewer NFf7 [1/2]**
>
> We thank you for your time and feedback and will incorporate your suggestions throughout the review into an updated manuscript. Below we first respond to the main points made in the review and then answer specific questions.
>
> ---
> > "I wonder how useful & interesting is the proposed setting, grounded in causal learning, compared to more standard MDP settings. I’m less familiar with this literature (but very familiar with MDP/RL literature), and it made it quite challenging to follow precisely the setting, algorithm and theoretical analysis."
>
> Policies considered in this paper are also referred to as dynamic treatment regimes (DTR) (Murphy, 2003) in the causal inference literature, which is commonly applied in medical decision-making. Different from standard MDP settings, DTR does not assume that the underlying environment satisfies the Markov property and the transition functions are generally different between different stages of intervention (action). A causal diagram encodes qualitative knowledge about the underlying and unobserved collection of causal mechanisms. Most of the existing methods in causal inference literature utilize causal diagrams in two ways. First, the agent could exploit the sparsity of the diagram to accelerate the online learning of optimal policies over a fixed set of actions. Second, structural constraints in the causal diagram allow the agent to evaluate the effects of policies using offline, observational data, with the presence of unobserved confounding bias. This paper studies a new dimension of causal reinforcement learning, namely, the online learning of the optimal combination of dynamic treatment regimes where the optimal set of actions is unknown. For instance, the agent might not have to intervene in all possible actions but only a subset of them. In the offline setting problem, this has been studied in Lee & Bareinboim, NeurIPS 2018 and 2020. One innovation in our method is to leverage the causal diagram so that the agent is able to evaluate the effects of a candidate policy from data collected by treatment regimes over different actions.
>
> ---
> >#### Q1. "Can you provide more detailed analysis of the space/time analysis for the algorithms CAUSAL-UCB* and the MINCOLLECTs?    The simulation results suggest this is limited to very small number of state/action variables; is this due to learning efficiency (in terms of #samples) or computational efficiency (in terms of memory/#operations)?  Could you include simulation results as a function of different factors (delta, #states/actions)?"
>
> IDENTIFY is a polynomial time algorithm with regard to the number of nodes and edges in the causal diagram. Since it calls subprocedure IDENTIFY at most $n^2$ time, MINCOLLECT is also a polynomial time algorithm.
>
> As for CAUSAL-UCB*, as stated in Line 283: “Nevertheless, solving polynomial optimization is NP-hard in general [15], which means that applying CAUSAL-UCB* is still computationally challenging.” To address the inherent challenge in this setting, we “introduce an alternative online algorithm,” called CAUSAL-TS*, “that is computationally feasible while achieving a similar asymptotic bound on the cumulative regret.” CAUSAL-TS* (Algorithm 3) allows us to reduce the original online learning problem by optimizing policies with mixed scopes in a fixed structural causal model (SCM). “There exist effective planning algorithms in finding optimal policies in 321 a structured environment provided with detailed parameterization of the underlying SCM [26, 23].” (Lines 320-321). Therefore, in general, we recommend using CAUSAL-TS* in practical applications.
>
> The computational efficiency of our method is equivalent to the state-of-art planning methods in structured environments. Therefore, it is not limited to a small number of states and actions. We performed additional simulations for Figure 2 where all variables take values in 5 possible outcomes. Simulation results showing cumulative regrets over $T = 5000$ episodes are provided below. One could see by inspection that the causal approach c-ts* consistently dominates other algorithms. We will include these additional simulations in the updated manuscript.
>
> |      | rct | ts | ts* | c-ts* |
> |--------|---------|---------|---------|---------|
> | Cumulative Regret | $5951$ |$5977$ | $5997$ | $2931$ |
>
> After all, we appreciate your feedback and would add that scaling these results for millions of variables is still non-trivial, and represent an open research direction.

---

> > ### Comment · Reviewer_NFf7 · 2022-08-09
> > **Comment following Author Response**
> >
> > Thank you for the additional information and clarification.
> > It is helpful to better understand the complexity of the algorithm.  Given it seems computationally feasible to address problems with at least dozens of variables, the simulation experiments remain very limited, especially given that for most diseases the causal effects do not operate so cleanly.
> > Overall, I have no objection to accepting this paper.  It is technically sound, clearly explained.
> > I find its potential for impact and significance to be lesser than other papers I have reviewed with similar / weaker reviewer scores.

---

> > > ### Author Response · Authors · 2022-08-10
> > > **Thank you for the response**
> > >
> > > We appreciate the reviewer’s response and will update the manuscript accordingly to highlight the main contributions of this paper. Here we would like to respectfully provide further contexts for the impact and significance of this work. First, we agree that there are many important challenges in RL, and one of them is policy learning in high-dimensional domains. In this paper, however, we investigate another challenge, arguably a crucial one as well: the theoretical analysis of online learning over the combinatorial space of sequential policies. These policies are also referred to in the literature as dynamic treatment regimes (DTRs), a popular decision-making model in medical domains and statistics (e.g., Murphy 2003). We propose the first online algorithms that could achieve sublinear regrets while optimizing over a mixed collection of DTR policies associated with different state-action spaces. Further, the regret bound is a sublinear function of candidate DTRs’ maximal cardinality of state-action space. This result answers the question posed in (30, Lee & Bareinboim, 2020), and now we have a precise quantification of how simplifying mixed scopes could accelerate future learning processes.

---

> ### Author Response · Authors · 2022-08-02
> **Response to Reviewer NFf7 [2/2]**
>
> ---
> >#### Q2. "Considering Experiment 2, only blood pressure is considered as a variable (not even age, gender and other well-known contributing factors).  Is this because the method is limited to such small domain?  Or because it necessitates strong assumptions?"
>
> The causal diagram in Experiment 2 is part of the literature and is introduced in (30, Lee, Correa & Bareinboim, 2019). Indeed, our method is not limited to specific causal diagrams with small domains but is applicable to any causal diagram with arbitrary structure. It reduces the online reinforcement learning (RL) problem into learning an optimal policy with mixed scope in a fixed SCM compatible with the provided diagram. We provide in our response to Q1 additional simulation results for SCMs with higher cardinalities of observed domains. We will also include additional simulations in the updated manuscript.
>
> ---
> >”The empirical results show an improvement for the proposed c-ts* method, compared to Thompson sampling alternatives. However the gap seems relatively small and both approaches seem sub-linear. How do you know if the improvement of c-ts* is significant? Is the results really meaningful?”
>
> We provide in our response to Q1 additional simulation results for SCMs with higher cardinalities of observed domains. One can see by inspection that the cumulative regret of the standard Thompson sampling becomes twice as large as the regret of c-ts*. This gap increases as the dimension of observed domains grows. We believe that this also supports our proposed Causal-TS* approach.
>
> ---
> >“The paper does not really discuss technical limitations of the work, except where it serves to motivate the proposed approach.”
>
> We explicitly stated the theoretical assumptions that are required for the proposed algorithms. For instance, “Throughout this paper, we assume that endogenous variables V are discrete and finite, while exogenous variables U could take any (continuous) value. Distributions $P(V)$ and $P(V | do(\pi))$ are thus categorical probability measures.” (Lines 128 -129) We also assume that the agent has access to “ a causal diagram associated with the underlying structural causal model (SCM)” (Line 59). These assumptions delineate the technical limitations of the work. Having said that, we will highlight and improve the manuscript to clarify these points further. Thanks for the suggestions.
>
>
> ---
> > “There could be a deeper discussion of the ethical aspects of this work, given the proposed application to selecting optimal treatment regime, and the example on patients with cardiovascular disease.”
>
> This paper investigates the online reinforcement learning (RL) of optimal policies with mixed scopes in an unknown structural causal model (SCM), provided with a causal diagram encoding qualitative knowledge about the underlying model. Such a system might be applicable to finding the optimal combination of dynamic treatment regimes for new diseases and discussed in (Murphy, 2003)  and many other applications that follow.  An immediate positive impact of this work is that the proposed method may help design novel adaptive experimental procedures to accelerate the learning of optimal treatment regimes. Such a procedure will consistently outperform existing randomized trial designs regarding the experimentation cost, which assigns patients to all candidate treatment regimes with equal chances. For instance, we can see by inspection in Fig. 3(a) that our proposed algorithm Causal-TS* (c-ts*) demonstrates significant improvement over fully randomized trials (rct) in the cumulative regret. On the other hand, online RL is still an idealized learning setting and may not necessarily reflect practical challenges in real-world clinical trials. For instance,  the patient’s response to the treatment is not immediate but only revealed a few months after the randomization. Further investigation is required to study RL algorithms' application in clinical trials.

---

### Official Review · Reviewer_jKpZ · 2022-07-11

**Rating:** 7
**Confidence:** 2
**Soundness:** 3 good
**Presentation:** 3 good
**Contribution:** 3 good

**Summary:**

The paper considers regret-minimizing algorithms for selecting an optimal treatment regime from a policy space characterized with mixed state-action scopes.

More precisely, the authors focus on a structural causal model, motivated by medical applications, where they present two algorithms: a UCB-based one, which however requires intractable planning, and Thompson sampling relaxation, which appears to be “less intractable” (i.e., the computational issue is about finding a near-optimal policy on a fixed MDP).

They show improvements compared to a naive baseline as well as theoretically over naive UCB.

**Questions:**

None of relevance, other than the computational tractability discussed above

**Limitations:**

Discusses above

**Strengths And Weaknesses:**

I am not an expert of the model assumed here, so I can only give high-level comments.
The authors use some standard technology (e.g., UCB, Thompson sampling) in the structural causal model. To be clear, both UCB and Thompson sampling need to be adapted, so there is some novelty in the work. They provide also a limited empirical evaluation. I suppose that the empirical evaluation needs to be restricted to small works since it might be too expensive to run these algorithm large scale.
Overall, the work is in an interesting positions, towards a possible application, all the while doing some theoretical contribution.
However, I am not familiar enough with the the causal literature to strongly support the work here.

The authors should discuss the computational consideration of finding a near-optimal policy on a fixed MDP in a better fashion. As I understand, even the Thomspon sampling-based algorithm could be intractable. This is of course strictly a limitation of planning and not of the proposal here, but it should nonetheless be discussed in more depth.

---

> ### Author Response · Authors · 2022-08-02
> **Response to Reviewer jKpZ**
>
> We thank the reviewer for the thoughtful feedback. We are glad that the reviewer found our paper  ‘interesting’ and that it has ‘some theoretical contributions.’ We have addressed the comments regarding the computational tractability below.
>
> ---
> >#### Q1. "I suppose that the empirical evaluation needs to be restricted to small works since it might be too expensive to run these algorithms large scale. … The authors should discuss the computational consideration of finding a near-optimal policy on a fixed MDP in a better fashion. As I understand, even the Thomspon sampling-based algorithm could be intractable. This is of course strictly a limitation of planning and not of the proposal here, but it should nonetheless be discussed in more depth."
>
> The main contribution of this paper is the first online learning algorithms that could optimize candidate policies with different state-action spaces with sublinear regret in an unknown structural causal model (SCM), or when the collection of underlying structural functions F and probability over exogenous variables P(u) are unknown. Our results reduce the online learning problem to finding an optimal solution for a given SCM. In many cases of interest, effective planning methods for optimizing policies in SCMs do exist (23, Koller & Milch 2003; 26, Lauritzen & Nilsson). These algorithms could be seen as generalizations of Bellman's equation in a fixed MDP while exploiting the sparsity in the causal diagram. When domains of observed variables are high-dimensional, solving for an exact solution could be computationally  challenging. In this case, effective approximate methods incorporating additional parametric assumptions could be pursued. Investigating and extending these methods is an ongoing subject of research (Guestrin, 2013; Liu & Ihler, 2012, 2013; Kumar, Zilberstein & Toussaint, 2015). We will add a note on the paper to reflect this discussion; thank you.
>
> ---
> >#### Q2. “However, I am not familiar enough with the causal literature to strongly support the work here.”
>
> We would like to further contextualize our work. There exist identification methods in causal inference literature that evaluate the effects of policies with mixed scopes from the combination of offline data and structural constraints encoded in a causal diagram (Pearl, 1995). However, these methods all focus on offline settings where the agent could not directly evaluate candidate policies in the actual system. One of the main challenges in designing an effective online algorithm is to compute the upper confidence bounds using data collected from policies with different state-action scopes. We address this challenge by proposing a novel parametric family of canonical SCMs (Theorem 2) that could represent the expected rewards of all candidate policies in the given causal diagram. This canonical family of SCMs could also be useful for offline learning, especially when the expected rewards are not uniquely identifiable from data due to the presence of unobserved confounding. Possible applications include partial identification (Kallus & Zhou, 2018, 2020; Namkoong, Keramati, Yadlowsky & Brunskill 2020), where the agent attempts to obtain bounds over the expected rewards from observational data.

---

### Meta-Review · Area_Chair_9rxc · 2022-08-26

**Recommendation:** Accept
**Confidence:** Less certain

**Metareview:**

The reviewers appreciated the direction of the work, but they all had rather low confidence despite each one being a foremost expert in RL, which may indicate a potential mismatch of interest with NeurIPS as a venue. They also found the potential applications limited, especially given the great amount of formalization. Nonetheless, they judged the developments on this new problem insightful and having the potential to inspire further work in the area and also found no reason to reject. In view of this, I recommend acceptance as I think the potential benefits outweigh the concerns of fit or practical relevance, and given there are no concerns regarding validity.

**Award:**

No

---

### Decision · Program_Chairs · 2022-09-14

Accept